# *Dnmt3a* knockout in excitatory neurons impairs postnatal synapse maturation and increases the repressive histone modification H3K27me3

Junhao Li[1†], Antonio Pinto-Duarte[2†‡], Mark Zander[3§], Michael S Cuoco[4], Chi-Yu Lai[2], Julia Osteen[2], Linjing Fang[5#], Chongyuan Luo[3,6¶], Jacinta D Lucero[2], Rosa Gomez-Castanon[3], Joseph R Nery[3,6], Isai Silva-Garcia[2], Yan Pang[2], Terrence J Sejnowski[2], Susan B Powell[7], Joseph R Ecker[3*], Eran A Mukamel[1*], M Margarita Behrens[2,7*]

[1]Department of Cognitive Science, University of California, San Diego, La Jolla, United States; [2]Computational Neurobiology Laboratory, Salk Institute for Biological Studies, La Jolla, United States; [3]Genomic Analysis Laboratory, Salk Institute for Biological Studies, La Jolla, United States; [4]Bioinformatics and Systems Biology Graduate Program, University of California, San Diego, La Jolla, United States; [5]Waitt Advanced Biophotonics Core, Salk Institute for Biological Studies, La Jolla, United States; [6]Howard Hughes Medical Institute, Salk Institute for Biological Studies, La Jolla, United States; [7]Department of Psychiatry, University of California, San Diego, La Jolla, United States

*For correspondence:
ecker@salk.edu (JRE);
emukamel@ucsd.edu (EAM);
mbehrens@salk.edu
(MMargaritaB)

[†]These authors contributed equally to this work

Present address: [‡]Ionis Pharmaceuticals, Carlsbad, United States; [§]Waksman Institute of Microbiology, Rutgers, The State University of New Jersey, Piscataway, United States; [#]Division of Biology and Biological Engineering, California Institute of Technology, Pasadena, United States; [¶]Department of Human Genetics, University of California, Los Angeles, Los Angeles, United States

Competing interest: The authors declare that no competing interests exist.

**Abstract** Two epigenetic pathways of transcriptional repression, DNA methylation and polycomb repressive complex 2 (PRC2), are known to regulate neuronal development and function. However, their respective contributions to brain maturation are unknown. We found that conditional loss of the de novo DNA methyltransferase *Dnmt3a* in mouse excitatory neurons altered expression of synapse-related genes, stunted synapse maturation, and impaired working memory and social interest. At the genomic level, loss of *Dnmt3a* abolished postnatal accumulation of CG and non-CG DNA methylation, leaving adult neurons with an unmethylated, fetal-like epigenomic pattern at ~222,000 genomic regions. The PRC2-associated histone modification, H3K27me3, increased at many of these sites. Our data support a dynamic interaction between two fundamental modes of epigenetic repression during postnatal maturation of excitatory neurons, which together confer robustness on neuronal regulation.

## Editor's evaluation

In this manuscript the authors conditionally knock out the DNA methyltransferase Dnmt3a in developing excitatory cortical neurons to determine the consequences for chromatin regulation, gene expression, and neuron function. As expected they find widespread loss of DNA methylation but also an increase in histone methylation (H3K27me3) at many similar regions of the genome, which they propose may be a mechanism of functional compensation. Overall this study offers new insights into the gene regulatory and neuronal cellular functions of an important chromatin regulatory process.

## Introduction

Epigenetic modifications of DNA and chromatin-associated histone proteins establish and maintain the unique patterns of gene expression in maturing and adult neurons (*Kundakovic and Champagne, 2015*). Neuron development requires the reconfiguration of epigenetic modifications, including methylation of genomic cytosine (DNA methylation, or mC) (*Guo et al., 2014*; *Lister et al., 2013*; *Stroud et al., 2017*) as well as covalent histone modifications associated with active or repressed gene transcription (*Fagiolini et al., 2009*; *Putignano et al., 2007*). While mC primarily occurs at CG dinucleotides (mCG) in mammalian tissues, neurons also accumulate a substantial amount of non-CG methylation (mCH) during postnatal brain development in the first 2–3 weeks of life in mice and the first two decades in humans (*Lister et al., 2013*). Accumulation of mCH, and the gain of mCG at specific sites, depend on the activity of the de novo DNA methyltransferase DNMT3A (*Gabel et al., 2015*). In mice, the abundance of *Dnmt3a* mRNA and protein peaks during the second postnatal week (*Lister et al., 2013*; *Stroud et al., 2017*), a time of intense synaptogenesis and neuronal maturation. Despite evidence for a unique role of *Dnmt3a* and mCH in epigenetic regulation of developing neurons, the long-term consequences of *Dnmt3a*-mediated methylation on brain function remain largely unknown (*Stroud et al., 2017*).

One challenge in investigating the developmental role of *Dnmt3a* has been the lack of adequate animal models. Deleting *Dnmt3a* around embryonic day 7.5 (E7.5) driven by the *Nestin* promoter dramatically impaired neuromuscular and cognitive development and led to early death (*Nguyen et al., 2007*). This early loss of *Dnmt3a* specifically affects the expression of long genes with high levels of gene body mCA (*Boxer et al., 2020*; *Kinde et al., 2016*). By contrast, deletion of *Dnmt3a* starting around postnatal day 14 driven by the *Camk2a* promoter caused few behavioral or electrophysiological phenotypes (*Feng et al., 2010*), with only subtle alterations in learning and memory depending on genetic background (*Morris et al., 2014*). These results suggest that *Dnmt3a* may play a critical role during a specific time window between late gestation and early postnatal life. During these developmental stages, regulated gains and losses of DNA methylation throughout the genome establish unique epigenomic signatures of neuronal cell types (*Qian et al., 2017*; *Luo et al., 2017*; *Mo et al., 2015*).

To address the role of *Dnmt3a*-dependent epigenetic regulation in the functional maturation of cortical excitatory neurons, we created a mouse line using the *Neurod6* promoter (*Schwab et al., 2000*) (Nex-Cre) to delete exon 19 of *Dnmt3a* (*Okano et al., 1999*). In this conditional knockout (cKO), *Dnmt3a* is functionally ablated in excitatory neurons in the neocortex and hippocampus starting in mid-to-late gestation (embryonic day E13–15) (*Goebbels et al., 2006*). In *Dnmt3a* cKO animals, DNA methylation was substantially disrupted in excitatory neurons, leading to altered behavior and synaptic physiology without early life lethality or overt brain morphological alterations. We generated deep DNA methylome, transcriptome, and histone modification data in *Dnmt3a* cKO and control pyramidal cells of the frontal cortex (*Supplementary file 1*), enabling a detailed assessment of the molecular basis of neurophysiological and behavioral phenotypes. We found that the polycomb repressive complex 2 (PRC2)-associated chromatin modification, H3K27me3, increases during postnatal development following the loss of mCH DNA methylation in *Dnmt3a* cKO neurons primarily at sites where mCG is also depleted. Our data support a dynamic interaction between two fundamental modes of epigenetic repression in developing brain cells.

## Results

### *Dnmt3a* cKO in pyramidal neurons during mid-gestation specifically impairs working memory, social interest, and acoustic startle

Previous studies of *Dnmt3a* KO mice yielded results ranging from little cognitive or health effects (*Feng et al., 2010*; *Morris et al., 2014*) to profound impairment and lethality (*Dura et al., 2022*; *Nguyen et al., 2007*). These studies suggest that the developmental timing and cell type of *Dnmt3a* loss may determine the extent of subsequent phenotypes. Here, we took a targeted approach by functionally ablating *Dnmt3a* in cortical pyramidal cells starting during mid-gestation (*Figure 1A*). We took advantage of the developmental onset of *Neurod6* expression between embryonic day E11 and E13, after the onset of *Nestin* expression (*Thompson et al., 2014*) but well before the major postnatal wave of reprogramming of the neuronal DNA methylome (*Figure 1—figure supplement*

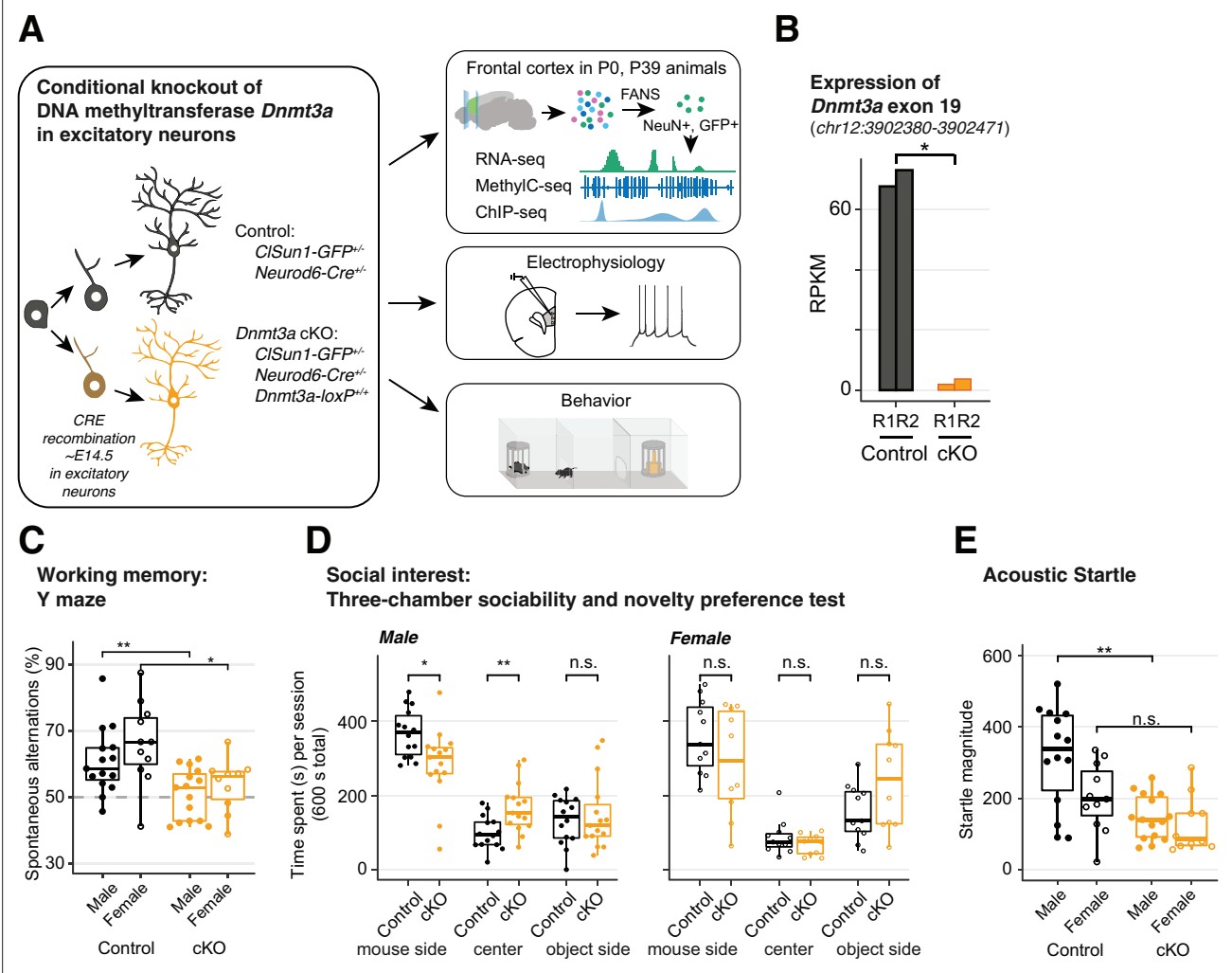

**Figure 1.** *Dnmt3a* conditional knockout (cKO) in cortical pyramidal neurons during mid-gestation impaired working memory, social interest, and acoustic startle responses. (**A**) An experimental model of the conditional loss of *Dnmt3a* in excitatory neurons. P0 and P39, postnatal day 0 and 39. FANS, fluorescence-activated nuclei sorting. (**B**) RNA-seq confirmation of the deletion of *Dnmt3a* exon 19 in P39 excitatory neurons. RPKM, reads per kilobase per million. R1/2, replicate 1/2. *, t-test p=0.014. (**C**) *Dnmt3a* cKO mice made fewer spontaneous alternations in the Y-maze test of working memory (Wilcoxon test, **, p=0.0079; *, p=0.011; n=15 male control, 15 male cKO, 11 female control, 10 female cKO). (**D**) Male *Dnmt3a* cKO mice spent less time interacting with an unfamiliar mouse, indicating reduced social interest (Wilcoxon test; *, p=0.01048; **p=0.006833; n=14 male control, 15 male cKO, 11 female control, 10 female cKO). (**E**) Male *Dnmt3a* cKO mice had decreased startle response to a 120 dB acoustic pulse (Wilcoxon test, **, p=0.0019; n.s., not significant; n=14 male control, 15 male cKO, 11 female control, 10 female cKO).

The online version of this article includes the following figure supplement(s) for figure 1:

**Figure supplement 1.** *Neurod6* starts to express between embryonic day E11 and E13.

**Figure supplement 2.** *Neurod6*-dependent Cre recombination occurred only in excitatory neurons.

**Figure supplement 3.** *Dnmt3a* was disrupted on both the mRNA and protein levels in the *Dnmt3a* conditional knockout (cKO) excitatory neurons.

**Figure supplement 4.** The conditional ablation of *Dnmt3a* in pyramidal neurons did not significantly impair motor activity nor increased anxiety levels.

**Figure supplement 5.** *Dnmt3a* conditional knockout (cKO)-induced impairment of startle response was accompanied by increased prepulse inhibition (PPI), and the cKO did not affect fear memory.

*1*; *Lister et al., 2013*). We confirmed that *Neurod6*-dependent Cre recombination occurred only in excitatory neurons using the INTACT (isolation of nuclei tagged in specific cell types) (*Mo et al., 2015*) mouse (*Figure 1—figure supplement 2*). The deletion of *Dnmt3a* exon 19 was faithfully captured in cortical excitatory neurons of cKO animals (*Figure 1B* and *Figure 1—figure supplement 3A*), which was known to produce a deletion of 50 amino acids in the methyltransferase domain of the DNMT3A protein (*Lyko, 2018*). We also verified the reduction in DNMT3A protein in whole tissue extracts at

early postnatal time points (P5 and P13) when *Dnmt3a* mediated accumulation of mCH normally begins in the frontal cortex (*Lister et al., 2013*; *Figure 1—figure supplement 3B*). *Dnmt3a* cKO animals survived and bred normally, without overt morphological alterations in the brain (*Figure 1—figure supplement 3C*). We found no impairments in gross motor function in an open field test: cKO mice traveled more during the first 5 min of testing (*Figure 1—figure supplement 4A*) while performing fewer rearings associated with exploratory interest (*Figure 1—figure supplement 4B*). Moreover, cKO mice had no signs of increased anxiety-like behavior on three separate behavioral tests (*Figure 1—figure supplement 4C-E*), in contrast with the reported anxiogenic effects of *Dnmt3a* knockdown in the mPFC of adult mice (*Elliott et al., 2016*). The absence of major impairments in overall health, motor function, or anxiety-like behavior established a baseline for investigating the role of *Dnmt3a* in specific cognitive and social behaviors.

We focused on cognitive domains associated with neurodevelopmental illness, including working memory and sensorimotor gating (*Habib et al., 2019*) and social interest (*Dodell-Feder et al., 2015*). *Dnmt3a* cKO mice did not alternate spontaneously between the arms of a Y-maze (p=0.0079 for males, p=0.011 for females, *Figure 1C*), indicating impaired spatial working memory. Moreover, when tested in a three-chamber box in which one of the sides contained a novel mouse and the opposite a novel object, male *Dnmt3a* cKO animals spent less time in the former, opting, instead, to remain significantly longer in the center (empty compartment), which is suggestive of reduced exploration and social interest (*Figure 1D*, left panel; p=0.01048). Male *Dnmt3a* cKO mice also had significantly attenuated acoustic startle reflex (p=0.0019, *Figure 1E* and *Figure 1—figure supplement 5A*). We observed increased prepulse inhibition (PPI) in male *Dnmt3a* cKO mice, but this may be driven by the reduced startle reflex (*Figure 1—figure supplement 5B*, p<0.05). It is noteworthy that the observed deficits in startle response were not due to impaired hearing, since *Dnmt3a* cKO mice displayed intact hearing in tests of PPI and fear conditioning.

To test whether these deficits in specific neurocognitive domains reflect generalized impairment in brain function, we assessed long-term memory using a fear conditioning paradigm. There were no significant differences between *Dnmt3a* cKO and control male mice in acquisition or recall of fear memory following re-exposure to the context or conditioned stimulus after 24–48 hr (*Figure 1—figure supplement 5C-E*), or in extinction (*Figure 1—figure supplement 5F*). Altogether, these behavioral results indicate that *Dnmt3a* cKO in excitatory neurons impairs working memory, social interest, and acoustic startle, without generalized cognitive disruption.

## Loss of *Dnmt3a* impairs synapse maturation and attenuates neuronal excitability

To test the impact of *Dnmt3a* cKO on dendritic morphology, we quantified the number and structure of 1278 DiI-labeled dendritic spines (NexCre/C57: n=701 from 5 mice; cKO: n=577 from 4 mice) of layer 2 pyramidal neurons of the mouse prelimbic region (~2 mm anterior to Bregma) (*Figure 2A*; Materials and methods), a region critical for working memory (*Yang et al., 2014*) and social approach behavior (*Lee et al., 2016*). While the overall density of dendritic spines was equivalent in control and *Dnmt3a* cKO neurons (*Figure 2B*), the spines were significantly longer (mean length 2.219±0.052 μm in cKO, 1.852±0.034 μm in control) and narrower (mean width 0.453±0.008 μm in cKO, 0.519±0.008 μm in control) in *Dnmt3a* cKO neurons (*Figure 2C*, KS test p<0.001; *Figure 2—figure supplement 1*). Consistent with this, a larger proportion of spines in *Dnmt3a* cKO mice were classified as immature filopodia, and fewer were mushroom-shaped mature spines (*Figure 2D*) according to pre-established morphometric criteria (see Materials and methods). The proportion of spines with other morphologies, including branched spines with more than one neck (data not shown), was not significantly different between genotypes (*Figure 2D*). These data indicate a role for *Dnmt3a* in dendritic spine maturation.

To test how the loss of *Dnmt3a* and subsequent stunting of spine maturation affect intrinsic neuronal excitability and synapse sensitivity, we next performed patch-clamp experiments in visually identified layer 2 pyramidal neurons from the prelimbic region (*Figure 2E*). Whole-cell current-clamp recordings showed that *Dnmt3a* cKO neurons (n=22 cells from 11 mice) required greater current injections than control (n=17 cells from 12 mice) to trigger an action potential (higher rheobase, t-test p=0.0042, *Figure 2F*), though there was no difference in membrane potential at the firing threshold (*Figure 2—figure supplement 2A*). *Dnmt3a* cKO neurons also produced fewer spikes in

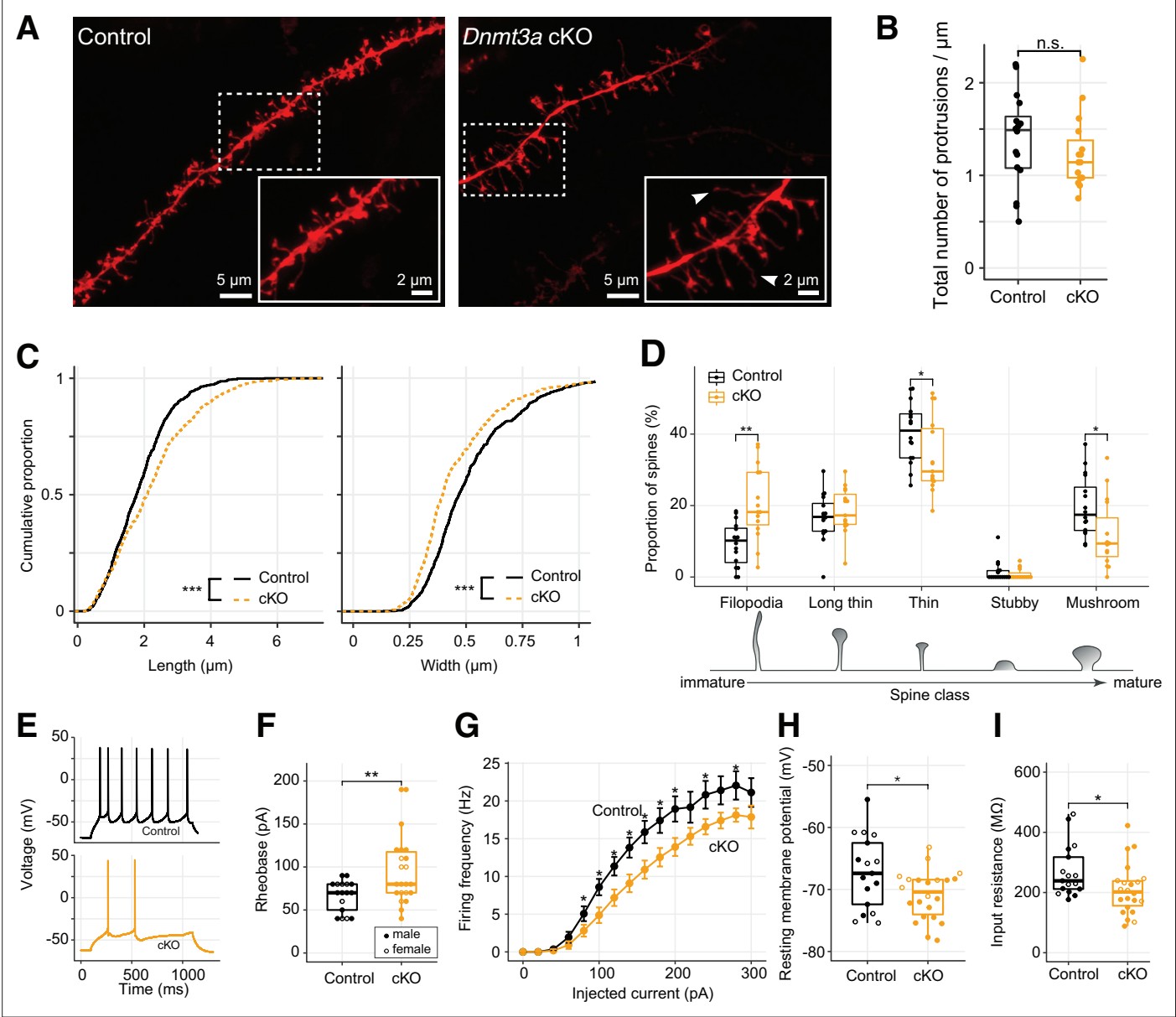

**Figure 2.** Immature spine morphology and reduced excitability of layer 2 excitatory neurons following *Dnmt3a* conditional knockout (cKO). (**A**) Example dendritic segments of layer 2 pyramidal neurons in the prelimbic region labeled with DiI and visualized using a 63× objective coupled to an Airyscan confocal microscope. Arrowheads show filopodia, which were more abundant in *Dnmt3a* cKO mice. (**B**) The density of membrane protrusions was unchanged in the *Dnmt3a* cKO (Wilcoxon test, n.s., not significant). (**C**) Membrane protrusions were significantly longer and narrower in the *Dnmt3a* cKO (KS test, p<0.001). (**D**) More spines were classified as immature filopodia, and fewer as mature mushroom-shaped spines with large postsynaptic densities in the *Dnmt3a* cKO (Wilcoxon test, **, p=0.0015; *, p=0.046 and 0.011 for thin and mushroom, respectively). (**E**) Example whole-cell patch-clamp recordings from prelimbic layer 2 pyramidal neurons following 60 pA current injections. (**F**) The median rheobase (i.e. the minimal current necessary to elicit an action potential) was significantly higher in the *Dnmt3a* cKO (t-test, **, p=0.0042). (**G**) Action potential frequency vs. injected current (mean ± SEM) showed reduced excitability in *Dnmt3a* cKO (Wilcoxon test, *, p<0.05). (**H**) and (**I**) *Dnmt3a* cKO neurons were slightly hyperpolarized at V_rest when compared to control (Wilcoxon test, *, p=0.049) and had lower membrane resistance (Wilcoxon test, *, p=0.023).

The online version of this article includes the following figure supplement(s) for figure 2:

**Figure supplement 1.** The membrane protrusions in the *Dnmt3a* conditional knockout (cKO) showed longer dendritic spines and narrower heads.

**Figure supplement 2.** The conditional ablation of *Dnmt3a* in pyramidal neurons did not significantly alter the membrane potential threshold for action potential generation, the mean amplitude, or frequency of miniature excitatory postsynaptic events.

response to injected current (*Figure 2G*). These neurons were slightly hyperpolarized at rest (mean –70.90±0.8 mV in cKO, n=22 cells from 11 mice vs. –67.22±1.4 mV in control, n=17 cells from 12 mice, Wilcoxon test p=0.049, *Figure 2H*), which could reflect differential expression of ion channels at the plasma membrane. Consistent with this, *Dnmt3a* cKO neurons had a lower input resistance (Wilcoxon test p=0.023, *Figure 2I*), suggesting increased expression of functional transmembrane ion channels. Whole-cell voltage-clamp recordings of miniature excitatory postsynaptic currents (mEPSCs) showed slight, yet significant, increased amplitude variability in *Dnmt3a* cKO mice (*Figure 2—figure supplement 2*, 8.77±0.32 pA in cKO, n=8 cells from 4 mice vs. 8.67±0.089 pA in control, n=8 cells from 5 mice, F-test, p=0.0032), consistent with disruption at postsynaptic sites. However, we found no alteration in the mean amplitude (*Figure 2—figure supplement 2B*) or frequency (*Figure 2—figure supplement 2C*) of mEPSCs recorded at the soma.

## Altered gene expression in *Dnmt3a* cKO excitatory neurons

To investigate the impact of epigenetic disruption on gene expression, we compared the transcriptomes of cKO and control excitatory neuron nuclei in mature mice (postnatal day 39) (*Supplementary file 2*). We isolated nuclei from excitatory neurons in frontal cortex by backcrossing the *Dnmt3a* cKO animals into the INTACT mouse (*Mo et al., 2015*) on a C57BL/6J background, followed by fluorescence-activated nuclei sorting (FANS) and RNA-seq. Although sorted nuclei contain only a subset of the cell's total mRNA and are enriched in immature transcripts, nuclear RNA-seq is nevertheless a quantitatively accurate assay of gene expression that is robust with respect to neural activity-induced transcription (*Bakken et al., 2018*; *Lacar et al., 2016*). All of the molecular assays were applied to at least two independent biological samples, with each sample being derived from a pool of tissue from 1 or 2 mice (*Supplementary file 1*). Nuclear RNA abundance was highly consistent across independent replicates within the same group (Spearman correlation r=0.93–0.94, *Figure 3—figure supplement 1A*). Given the repressive role of DNA methylation (mCG and mCH) in regulating gene expression in neurons (*Guo et al., 2014*; *Lister et al., 2013*), we expected to find increased gene expression in the cKO. Consistent with this, we detected 46 differentially expressed (DE) genes with higher expression in the cKO (false discovery rate [FDR] < 0.05, *Figure 3A–B* and *Figure 3—figure supplement 1B*, *Supplementary file 2*). We also detected significantly lower expression of 24 genes in cKO neurons (*Figure 3A*). Several of the DE genes had annotated roles in dendrite morphogenesis (*Elavl4*, *Hecw2*, *Ptprd*), as well as in the regulation of $Na^+$ (*Hecw2*, *Scn3b*) and $Ca^{2+}$ levels (*Cacnb3*) (*Figure 3B*). Early life experiences were shown to alter the expression of *Dnmt3a* and hence the DNA methylation levels in some transposable elements (TEs) in the hippocampus (*Bedrosian et al., 2018*). We examined the expression of transposon families or classes in our data but found no significant changes between cKO and control in the cortical excitatory neurons (two-sample t-test, FDR > 0.05; *Figure 3—figure supplement 2*).

## *Dnmt3a* cKO abolishes postnatal DNA methylation

Deletion of *Dnmt3a* during mid-gestation should disrupt the subsequent gain of DNA methylation at specific genomic sites during development (*Qian et al., 2017*; *Lister et al., 2013*), without affecting sites that maintain or lose methylation after E14.5. Using single base resolution, whole-genome MethylC-seq (*Lister et al., 2008*) in biological replicates with strong consistency (*Figure 3—figure supplement 3A*), we confirmed that non-CG DNA methylation (mCH) in excitatory neurons is absent at birth (<0.1% of all CH sites at P0), and accumulates by postnatal day 39 (1.98% at P39) (*Lister et al., 2013*). The cKO all but eliminated mCH (<0.1% at P0 and P39) (*Figure 3C* and *Figure 3—figure supplement 3B*). While mCH increases in neurons during postnatal life, the genome-wide level of mCG in the brain remains high throughout the lifespan (*Lister et al., 2013*). We found that the genome-wide mCG level was 12.5% lower in mature (P39) cKO neurons (60.1% in cKO vs. 72.6% in control). There was no difference in mCG in newborn mice (P0, 73.1% in both cKO and control, *Figure 3C*). mCG at P39 was reduced in 92.2% of all genomic bins (10 kb resolution), and was significantly lower in introns, 3' UTR, and intergenic regions (*Figure 3—figure supplement 3C*). The reduction in mCG was strongly correlated with reduced mCH (Spearman correlation r=0.805, p<10⁻³, *Figure 3—figure supplement 3D*). These data support a role for *Dnmt3a* in postnatal de novo CG and CH DNA methylation across genomic compartments (*Lister et al., 2013*; *Stroud et al., 2017*).

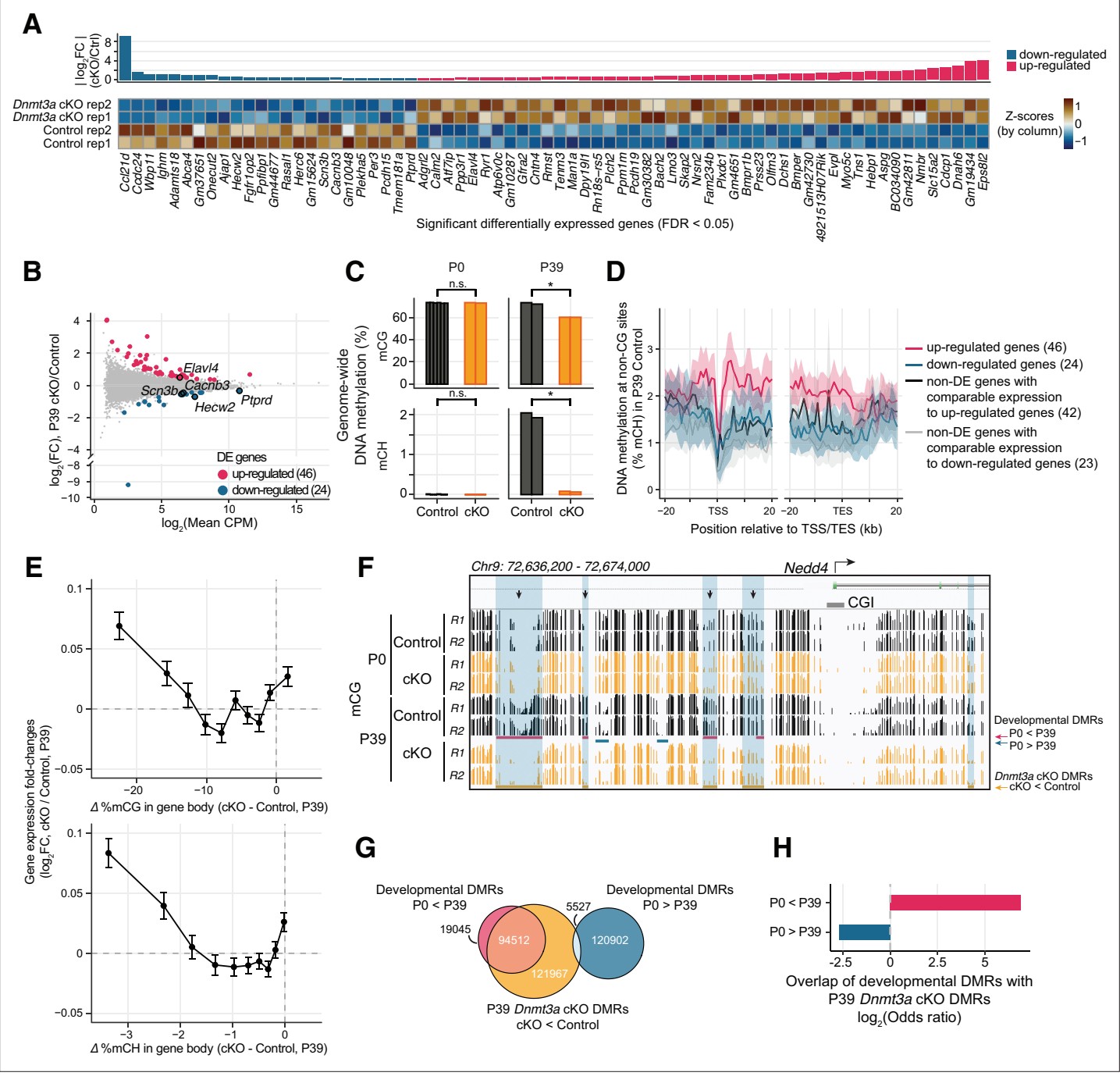

**Figure 3.** Loss of *Dnmt3a* leaves thousands of genomic regions in a fetal-like demethylated state. (**A**) 70 genes were differentially expressed (DE) (false discovery rate [FDR] < 0.05) in P39 pyramidal neurons in *Dnmt3a* conditional knockout (cKO) vs. control. Top, fold-change (FC) between cKO and control; bottom, heatmap showing normalized expression of the DE genes in each sample. Z-scores were computed using mRNA counts per million (CPM) for each DE gene. (**B**) Differential gene expression in control vs. *Dnmt3a* cKO excitatory neurons at P39. Significant up-regulated and down-regulated DE genes are shown in red and blue, respectively. Differentially expressed (DE) genes associated with dendrite morphogenesis (*Elavl4*, *Hecw2*, *Ptprd*), and regulation of Na+ (*Hecw2*, *Scn3b*) and Ca2+ levels (*Cacnb3*) are labeled. (**C**) Non-CG DNA methylation (mCH) is eliminated, and mCG is reduced, in P39 *Dnmt3a* cKO pyramidal cells, while mCG and mCH levels are not changed in P0 (t-test: *, p<0.05; n.s., not significant). P0 and P39, postnatal days 0 and 39, respectively. Each bar represents the methylation level in one replicate. (**D**) Non-CG DNA methylation (mCH) in P39 pyramidal cells in control samples, in 1 kb bins in the flanking region around the transcription start (TSS) and end site (TES) of DE genes and non-DE genes with matched expression levels. The lines denote the means across genes in each gene set, and the shared areas represent the 95% confidence intervals of the means. (**E**) The difference in gene body methylation vs. fold-change of gene expression between P39 *Dnmt3a* cKO and control. The plots show mean ± SEM gene expression fold-change for genes in 10 non-overlapping bins (deciles of mC difference). (**F**) The *Nedd4* promoter locus contains five

*Figure 3 continued on next page*

*Figure 3 continued*

differentially methylated regions (DMRs, yellow horizontal rectangles and shaded in blue boxes) with naive, fetal-like mCG in P39 *Dnmt3a* cKO. Ticks show mCG at CG sites. Four out of the five P39 *Dnmt3a* cKO DMRs overlapping developmental gain-of-methylation DMRs (red horizontal rectangles) are marked with arrows. CGI, CpG island. R1 and R2, replicates 1 and 2. (**G**) Overlap of P39 *Dnmt3a* cKO DMRs and developmental DMRs. (**H**) P39 *Dnmt3a* cKO hypo-DMRs are significantly enriched (depleted) in DMRs that normally gain (lose) methylation during development (Fisher's test, p<0.05).

The online version of this article includes the following figure supplement(s) for figure 3:

**Figure supplement 1.** RNA-seq data showed transcriptomic disruption in P39 *Dnmt3a* conditional knockout (cKO) pyramidal neurons.

**Figure supplement 2.** The expression of transposable elements (TEs) was not affected by *Dnmt3a* conditional knockout (cKO).

**Figure supplement 3.** Genome-wide reduction of DNA methylation was observed in *Dnmt3a* conditional knockout (cKO).

**Figure supplement 4.** Reduction of DNA methylation cannot fully explain the disruption in the transcriptome after *Dnmt3a* conditional knockout (cKO).

**Figure supplement 5.** Differentially methylated regions (DMRs) in P39 *Dnmt3a* conditional knockout (cKO) were associated with regulatory regions in enhancers and repressed chromatin.

## Reduced DNA methylation does not fully explain altered transcription in *Dnmt3a* cKO

We investigated whether the altered gene expression in *Dnmt3a* cKO neurons correlated with loss of DNA methylation at specific sites. We first analyzed DNA methylation around DE genes in mature neurons (P39). The simple model of DNA methylation as a repressive regulator of gene expression predicts that genes that lose the most mC should be most transcriptionally up-regulated in the cKO. Consistent with this, we found that mCH was strongly enriched in the gene body of up-regulated genes in control neurons (*Figure 3D* and *Figure 3—figure supplement 4A*). By contrast, genes with similar expression levels in the control neurons which were not transcriptionally up-regulated in the cKO (*Figure 3—figure supplement 4B*) had significantly lower gene body mCH (*Figure 3D*). Moreover, down-regulated genes had low gene body mCH. These data support a causal role for gene body mCH in repressing gene expression. The relatively lower mCH level in down-regulated genes could make them less sensitive to the loss of *Dnmt3a*. The dysregulation of their expression may be due to secondary effects subsequent to the direct loss of DNA methylation.

We also examined the pattern of mCG at the promoter and gene body of DE genes. In contrast with the pattern of mCH, mCG was not significantly different between DE and control genes (*Figure 3—figure supplement 4C*).

The difference in gene body methylation (cKO – control) was negatively correlated with gene expression changes, consistent with repressive regulation (*Gabel et al., 2015*; *Lavery et al., 2020*; *Figure 3E*). This correlation accounted for 0.46% of the variance of differential gene expression, whereas the total explainable variance ($R^2$ between biological replicates) was 1.30% (*Figure 3—figure supplement 4D*). The strength of the association between mCH and mRNA changes may be limited by the use of only two biological replicates in our dataset.

We next sought to determine if up- and down-regulated genes differ in ways that could explain their different responses to the loss of *Dnmt3a*. Up-regulated genes were on average longer than down-regulated genes (*Figure 3—figure supplement 4E*, Wilcoxon rank-sum test $p<10^{-4}$) and non-DE genes (p<0.01), consistent with the reported enrichment of mCA and MeCP2-dependent gene repression in long genes (*Boxer et al., 2020*; *Gabel et al., 2015*; *Kinde et al., 2016*). However, there was a broad distribution of gene lengths for both up- and down-regulated genes (*Lavery et al., 2020*).

In addition to promoters and gene bodies, distal regulatory elements such as enhancers are major sites of dynamic DNA methylation where epigenetic regulation can activate or repress the expression of genes over long genomic distances through 3D chromatin interactions (*Malik et al., 2014*). We investigated gene regulatory elements by identifying differentially methylated regions (DMRs) where mCG is altered in cKO compared to control neurons. We found a limited number of DMRs in newborn mice (P0: 1087 DMRs with lower, 164 with higher mCG in cKO; ≥30 difference in %mCG; FDR < 0.01). In mature neurons (P39), by contrast, we found 222,006 DMRs with substantially lower mCG in cKO compared with controls (*Figure 3—figure supplement 5A*; *Supplementary file 3*). Only 89 DMRs had ≥30 higher %mCG in cKO. To illustrate, we found five DMRs in an ~40 kb region around the promoter of the DE gene *Nedd4* (*Figure 3F*). Four of these DMRs were also unmethylated in excitatory neurons in newborn mice, showing that the loss of *Dnmt3a* blocked the developmental

gain of mCG at these sites. However, the density of P39 cKO DMRs around the DE genes was not significantly different in the up- and down-regulated genes and the non-DE genes (*Figure 3—figure supplement 5B*).

The majority of P39 cKO DMRs (68.0%) were distal (≥10 kb) from the annotated transcription start sites. These DMRs were significantly enriched in both active enhancers and repressed chromatin, suggesting they have a regulatory role (*Figure 3—figure supplement 5C*; see also below). We found 113,557 developmental DMRs that gain mCG between P0 and P39 in control neurons (≥30 difference in %mCG, FDR < 0.01). These DMRs strongly overlapped (83.2%) with the cKO DMRs in mature neurons (*Figure 3G–H*, *Supplementary file 3*). Moreover, the P39 cKO DMRs were enriched in DNA sequence motifs of multiple transcription factors (TFs) associated with neuronal differentiation, including *Rest*, *Lhx2*, *Pou3f2(Brn2)*, and *Pax6* (FDR < 0.05, *Figure 3—figure supplement 5D*, *Supplementary file 4*). Notably, the DMRs in *Dnmt3a* cKO neurons represent only a part of the global reduction in mCG that we observed throughout the genome. Indeed, we found that mCG is reduced by ~10% in all genomic compartments, even after excluding P39 DMRs (*Figure 3—figure supplement 3C*). These results suggest that *Dnmt3a* is essential for the methylation and subsequent repression of neuronal enhancers that are active during prenatal brain development.

## Increased PRC2-associated repressive histone modification H3K27me3 in *Dnmt3a* cKO

Given that the cKO loses DNA methylation throughout the genome, we were surprised that the expression level of many genes was not disrupted. We, therefore, explored other potential epigenetic regulators which could contribute to maintaining gene expression. To identify TFs and chromatin regulators with experimental evidence of binding at cis-regulatory regions of the DE genes, we performed Binding Analysis for Regulation of Transcription (BART) (*Wang et al., 2018*). Chromatin regulators associated with PRC2, including *Ezh2*, *Suz12*, *Eed,* and *Jarid2*, were among the top DNA binding proteins enriched near the promoters of both up- and down-regulated DE genes (*Figure 4A*). Several TFs associated with chromatin organization, including the histone deacetylase (*Hdac*) and demethylase (*Kdm*) families, and *Ctcf*, were also enriched (*Figure 4—figure supplement 1*, *Supplementary file 5*). These results suggest that *Dnmt3a* cKO impacts the chromatin landscape in excitatory neurons, potentially via altered PRC2 activity.

To experimentally address this, we performed chromatin immunoprecipitation sequencing (ChIP-seq) in excitatory neurons at embryonic day 14 (E14) and postnatal days 0 and 39 to measure trimethylation of histone H3 lysine 27 (H3K27me3), a repressive mark whose deposition is catalyzed by PRC2 and is important for transcriptional silencing of developmental genes. In P39 neurons, we also measured two histone modifications associated with active chromatin: H3K4me3 (trimethylation of histone H3 lysine 4, associated with promoters) and H3K27ac (acetylation of histone H3 lysine 27, associated with active promoters and enhancers) (*Heinz et al., 2015*). For each mark, we performed sequencing on two independent samples, each of which used pooled tissue from two mice. The active and repressive marks had positive and negative correlations with mRNA expression, respectively (*Figure 4—figure supplement 1B*). In addition, we noted regions where increased H3K27me3 concurred with the decreased DNA methylation. For example, at the *Mab21l2* locus (*Figure 4B*), we observed a 2.41-fold increase (*Figure 4C*) in H3K27me3 in P39 *Dnmt3a* cKO neurons, coinciding with the loss of CG methylation at multiple DMRs spanning the gene body and surrounding region. The *Mab21l2* gene was lowly expressed (TPM < 3) in both control and cKO neurons, consistent with a role for the gain of H3K27me3 in maintaining the repression of this gene.

Using a conservative strategy to call ChIP-seq peaks (*Zang et al., 2009*), we found that marks associated with transcriptional activity (H3K4me3 and H3K27ac) were largely conserved in the P39 cKO and control (*Figure 4D*). By contrast, we found 51.9% more H3K27me3 peaks in mature (P39) cKO than in control neurons (*Figure 4D*, *Supplementary file 6*). When we directly identified differentially modified (DM) regions, we found no DM for H3K4me3 and H3K27ac between the cKO and control in P39 neurons (*Figure 4—figure supplement 1C*). By contrast, we found 4040 regions with significantly increased H3K27me3 in the P39 cKO, covering ~31.05 MB of the genome (*Figure 4—figure supplement 1D*, FDR < 0.05, *Supplementary file 7*). Differential H3K27me3 appears late during brain maturation: only 3 DM regions were found in earlier development stages (E14 or P0; *Figure 4—figure supplement 1D*), and the signal differences of cKO and control did not show clear correlations

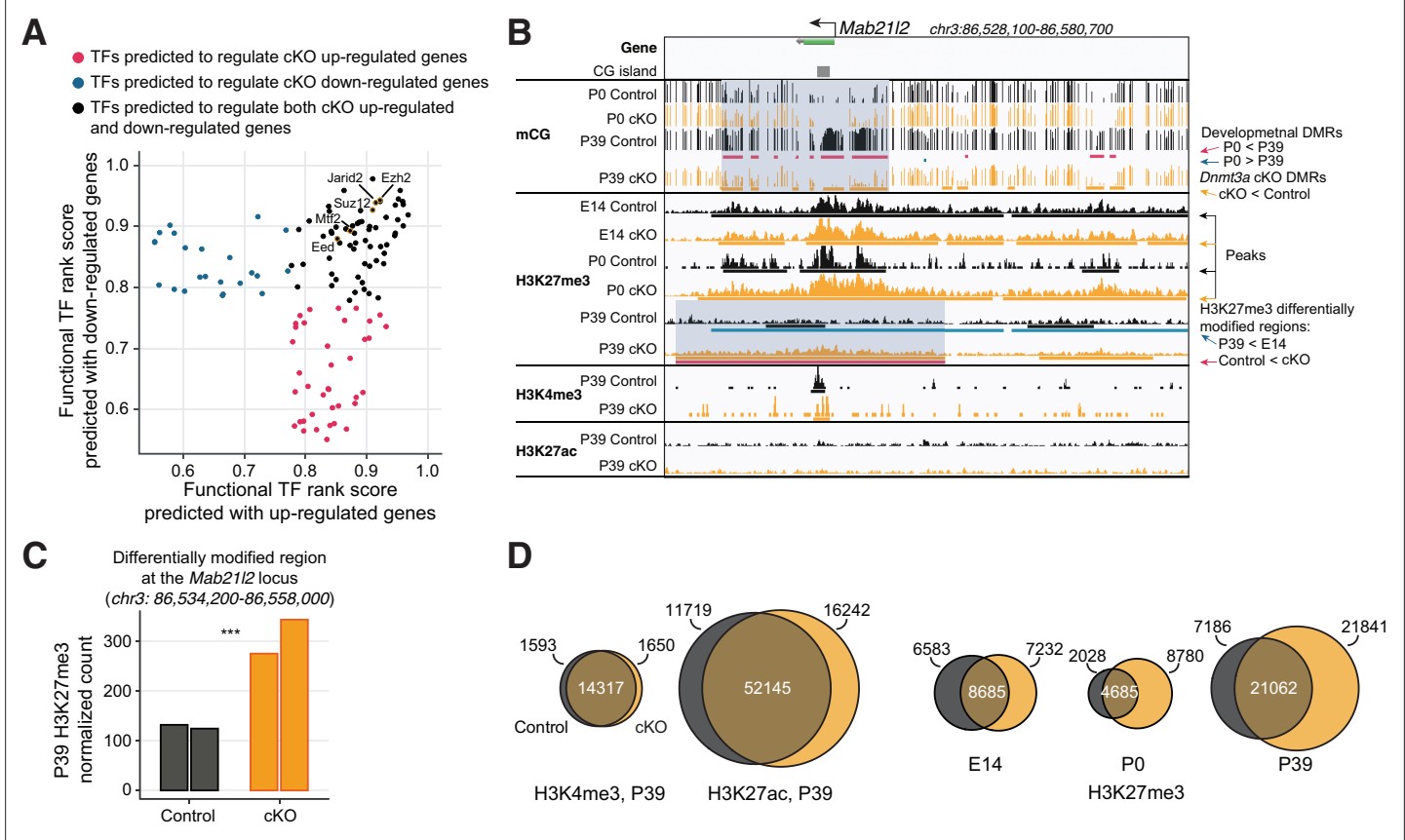

**Figure 4.** Polycomb repressive complex 2 (PRC2) associated histone modification H3K27me3 is up-regulated following the loss of DNA methylation. (**A**) Transcription factors (TFs) predicted to regulate P39 *Dnmt3a* conditional knockout (cKO) differentially expressed genes include many proteins associated with PRC2. The functional TF rank score was assigned by Binding Analysis of Regulation of Transcription (BART; *Wang et al., 2018*). PRC2-associated TFs are labeled and highlighted in gold circles. (**B**) Browser view of the *Mab21l2* locus, where increased H3K27me3 (differentially modified regions, bottom red bars and highlighted in blue shaded box) coincides with the loss of DNA methylation (*Dnmt3a* cKO DMRs, orange bars under the 'P39 cKO' track and highlighted in blue shaded box) in P39 *Dnmt3a* cKO. This region loses H3K27me3 during normal development in control pyramidal neurons (blue bars, P39 < E14). DMR, differentially methylated region; E14, embryonic day 14; P0 and P39, postnatal days 0 and 39. (**C**) Quantification of the increase in H3K27me3 chromatin immunoprecipitation sequencing (ChIP-seq) signal in each replicate at the H3K27me3 differentially modified region between P39 control and cKO at the *Mab21l2* locus shown in (**B**). Each bar shows the DEseq2 normalized counts in each replicate, and the triple asterisks denote a significant increase (false discovery rate [FDR] = 1.33e-4, fold-change=2.41). (**D**) Histone modification ChIP-seq peaks for active marks (H3K4me3, H3K27ac) are largely preserved in the *Dnmt3a* cKO, while repressive H3K27me3 peaks expand. The Venn diagrams denote numbers of peaks that overlap between cKO (yellow) and control (black) (numbers in the center), and numbers of peaks that are unique to one of the conditions (numbers on the edges).

The online version of this article includes the following figure supplement(s) for figure 4:

**Figure supplement 1.** Increased signal of the repressive histone mark H3K27me3 after *Dnmt3a* conditional knockout (cKO).

**Figure supplement 2.** The increased H3K27me signal in P39 was generally not observed in E14 or P0.

**Figure supplement 3.** H3K27me3 differentially modified (DM) regions in P39 *Dnmt3a* conditional knockout (cKO) were highly overlapped with differentially methylated regions (DMRs).

between P39 and earlier stages (*Figure 4—figure supplement 2*). These DM regions have a medium but non-zero level of H3K27me3 in the P39 control (higher than random shuffles across the whole genome but lower than random shuffles within the peak regions, *Figure 4—figure supplement 3A*), and hence presumably fine-tunable after *Dnmt3a* cKO. Genes associated with these DM regions were enriched in development-related functions (*Figure 4—figure supplement 3B*, *Supplementary file 8*). These results suggest that the increase of H3K27me3 in *Dnmt3a* cKO excitatory neurons occurred postnatally, following the major impact of the loss of *Dnmt3a* on neuronal DNA methylation.

The increase in H3K27me3 was closely associated with regions that lost CG DNA methylation in the *Dnmt3a* cKO. The majority (66.2%) of the regions marked by H3K27me3 in the cKO overlapped

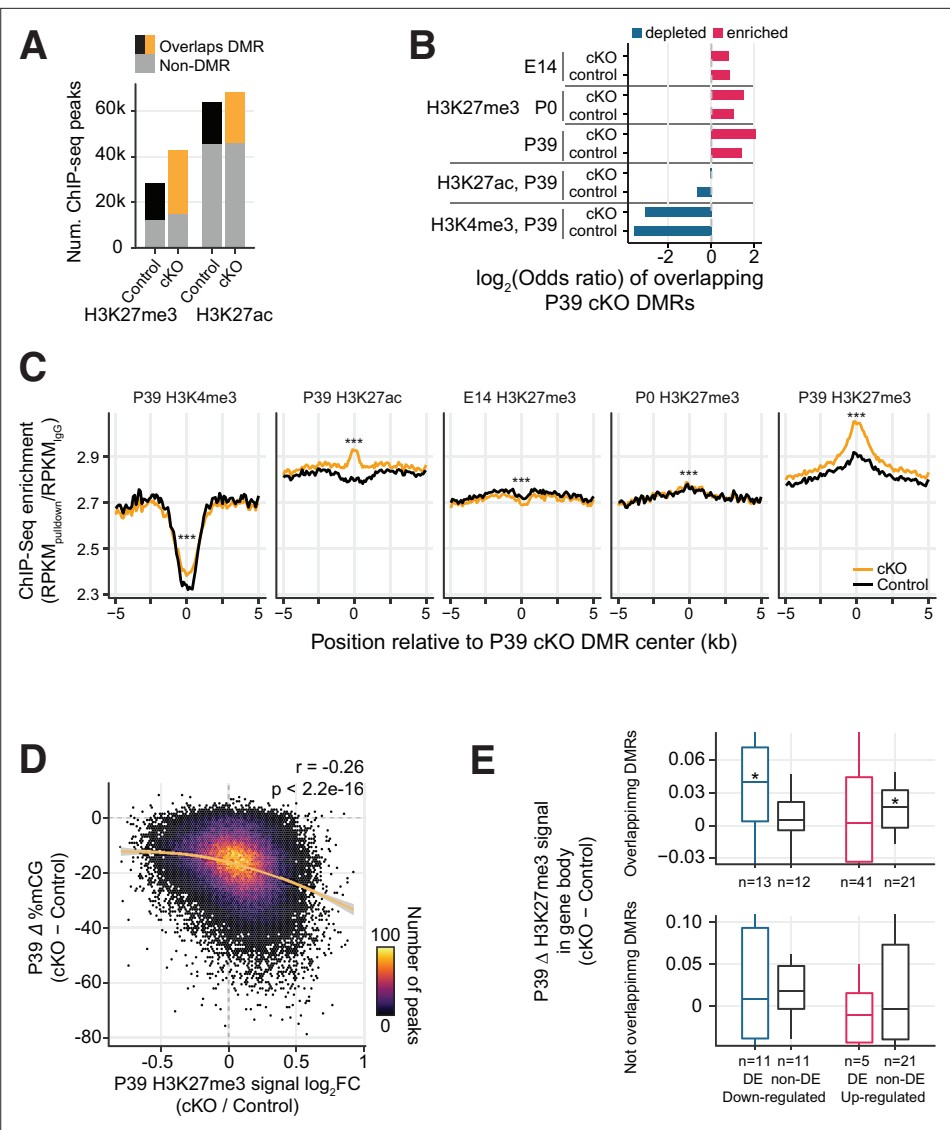

**Figure 5.** Increased H3K27me3 correlates with the loss of postnatal DNA methylation. (**A**) Most of the P39 H3K27me3 peaks (57.1% and 66.2% of control and conditional knockout [cKO] peaks), but only some of the P39 H3K27ac peaks (28.9% and 33.1%), overlap with P39 *Dnmt3a* cKO differentially methylated regions (DMRs). (**B**) Significant enrichment (red) or depletion (blue) of P39 *Dnmt3a* cKO DMRs in the histone modification chromatin immunoprecipitation sequencing (ChIP-seq) peaks (Fisher's test, p<0.05). E14, embryonic day 14; P0 and P39, postnatal days 0 and 39. (**C**) Histone modification ChIP-seq signal around the center of DMRs. RPKM, reads per kilobase per million. ***, Wilcoxon rank-sum test of the differences at the center, p<0.001. (**D**) Correlation of P39 H3K27me3 signal fold-changes and P39 CG methylation levels differences between *Dnmt3a* cKO and control in H3K27me ChIP-seq peaks. The smoothed line is fitted using a generalized additive model, and the shaded area shows the 95% confidence interval of the fit. r, Spearman correlation coefficient. (**E**) P39 *Dnmt3a* cKO down-regulated DE genes (false discovery rate [FDR] < 0.05) with overlapping P39 *Dnmt3a* cKO DMRs show small but significant increases of H3K27me3 in P39 cKO (upper panel). *, Wilcoxon rank-sum test against zero, p<0.05. No such differences were observed in differentially expressed (DE) genes that do not overlap with P39 cKO DMRs (bottom panel). H3K27me3 signal was calculated as the RPKM fold-change between H3K27me3 and IgG.

The online version of this article includes the following figure supplement(s) for figure 5:

**Figure supplement 1.** Differentially methylated regions (DMRs) are particularly unique in showing increased H3K27me3 signal.

**Figure supplement 2.** Relationships between gene expression, H3K27me3 signal and gene body mCH.

with P39 cKO DMRs (*Figure 5A*, Fisher's test p<1e-100). Likewise, P39 cKO DMRs were significantly enriched in peaks (22.4% overlapped H3K27me3 peaks in cKO, p<1e-100, *Figure 5B*) and DM regions of H3K27me3 (*Figure 4—figure supplement 3D*). The DM regions of H3K27me3 had significantly more overlaps with DMRs compared to the non-DM control regions (Fisher's exact test p<2.2e-16, OR: 2.70, *Figure 4—figure supplement 3C-D*). The DM regions also have larger decreases in both mCG and mCH when compared to non-DM regions (Wilcoxon rank-sum test p<0.0001, *Figure 4—figure supplement 3E*). Conversely, the DMRs were depleted in regions marked by H3K27ac (12.7%, p<1e-100) or H3K4me3 (0.98%, p=1.73e-149) (*Figure 5A–B*). Moreover, H3K27me3 was more abundant at the center of DMRs in cKO compared to control neurons at P39 (Δ=0.15 in the unit of fold enrichment vs. IgG, 5.04% increases, *Figure 5C*). There were much smaller changes of H3K27me3 at these DMRs in newborn (P0, Δ=0.0075, 0.27% changes) or fetal (E14, Δ=–0.028, –1.02% changes) neurons (*Figure 5C*), and the increases were not seen in randomly shuffled regions (*Figure 5—figure supplement 1A*). The signal of H3K4me3 and H3K27ac at the DMRs was also elevated in the cKO, to a lesser extent (H3K4me3: Δ=0.068, 2.92% changes; H3K27ac: Δ=0.13, 4.64% changes). Going beyond overlaps of regions, we found a quantitative association between the changes in DNA methylation and H3K27me3 in mature (P39) neurons (*Figure 5D*). At H3K27me3 peaks, the ChIP-seq signal intensity fold-change between cKO and control correlated with the loss of mCG in cKO (Spearman r=–0.26, p<2.2e-16). When accessing the H3K27me3 changes as a function of baseline H3K27me3 levels in the control across the genome tiled in 1 kb bins, we observed bigger H3K27me3 increases in bins with overlapping DMRs compared to bins without overlapping DMRs (Wilcoxon rank-sum test p<0.001 in each bin, *Figure 5—figure supplement 1B*). These results indicate that DMRs are particularly unique in showing increased H3K37me3 signal.

We further examined whether the increased H3K27me3 could account for the reduced expression of some genes in the *Dnmt3a* cKO neurons. We found that down-regulated genes that overlap DMRs and the non-DE genes selected with matched expressions of up-regulated DE genes that overlap DMRs showed a small but significant median increase of H3K27me3 in the cKO (Wilcoxon rank-sum test p-value < 0.05; *Figure 5E* upper panel). When we considered a larger set of DE genes with a more relaxed threshold (FDR < 0.2), we observed that down-regulated DE genes containing DMRs accumulate significantly more H3K27me3 than non-DE genes containing DMRs and up-regulated genes containing DMRs (Wilcoxon rank-sum test p=0.0029, *Figure 5—figure supplement 2A*). By contrast, up-regulated genes had no significant accumulation of H3K27me3 and were not significantly different from the control genes (*Figure 5—figure supplement 2A*). No such differences were observed between DE and non-DE genes without overlapping DMRs (*Figure 5E* lower panel and *Figure 5—figure supplement 2A*). These results suggest that the effect of H3K27me3 is likely specific to genes containing DMRs and the effect is stronger in the down-regulated DE genes, which may partially explain the fact that 24 genes were significantly down-regulated after the loss of repressive DNA methylation in the *Dnmt3a* cKO (*Figure 2A–B*).

We next analyzed how changes in H3K27me3 related to the loss of mCH. In all non-DE genes (FDR ≥ 0.05), we observed a larger increase of H3K27me3 signal in the gene body of genes that lost most mCH in the cKO (*Figure 5—figure supplement 2B*). Indeed, when we grouped all genes by the extent of the loss of mCH, we found that genes that lost the most mCH had a negative correlation between the changes in gene body H3K27me3 signal and gene expression fold-change. Such correlation was not observed in genes that did not lose mCH (*Figure 5—figure supplement 2C*). These results suggest that H3K27me3 may have a role in repressing expression specifically in regions that lose repressive non-CG DNA methylation.

## Developmental changes in H3K27me3 are not affected by *Dnmt3a* cKO

Our finding that *Dnmt3a* cKO disrupts the normal developmental gain of DNA methylation prompted us to ask whether the changes in H3K27me3 in the cKO are likewise associated with developmental regulation of H3K27me3. Indeed, our ChIP-seq data from E14, P0, and P39 excitatory neurons revealed striking developmental dynamics in H3K27me3. We identified 12,994 developmentally regulated H3K27me3 DM regions between E14 and P39, with a similar number of regions that gain (6774) and lose (6220) H3K27me3 (*Figure 6A* and *Figure 6—figure supplement 1A*, *Supplementary file 9*). We also examined DM regions at P0 vs. E14 and P0 vs. P39 (*Figure 6—figure supplement 1A*). However, due to greater biological variability at the perinatal time point, the two replicate ChIP-seq

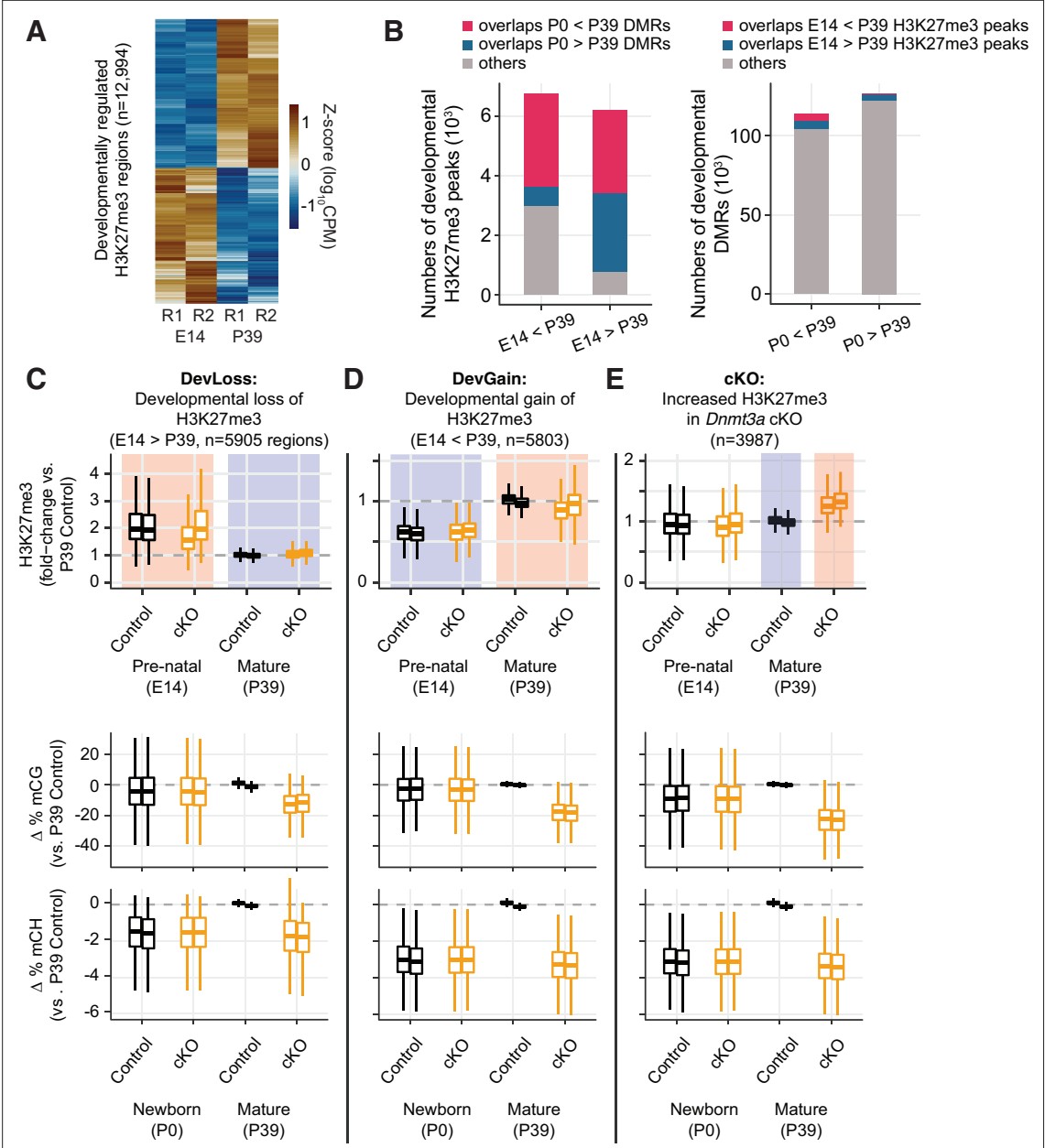

**Figure 6.** Developmental dynamics of H3K27me3. (**A**) Heatmap of developmentally regulated H3K27me3 regions in E14 and P39 control samples. CPM – counts per million; R1/2 – replicates. (**B**) Bar plots show the numbers of developmental differentially modified H3K27me3 regions (E14 vs. P39) that overlap developmental differentially methylated regions (DMRs) (P0 vs. P39, left panel), and the numbers of developmental DMRs that overlap developmental differentially modified H3K27me3 regions (right panel). (**C–E**) Normalized H3K27me3 signal (fold-changes compared to P39 control), and mCG, mCH differences (compared to the average of the two replicates from P39 control) in peaks that overlap with E14 vs. P39 developmental loss-of-H3K27me3 regions (**C**), developmental gain-of-H3K27me3 regions (**D**), or increased H3K27me3 in P39 *Dnmt3a* cKO (**E**).

The online version of this article includes the following figure supplement(s) for figure 6:

**Figure supplement 1.** Regions prone to alteration of H3K27me3 by *Dnmt3a* conditional knockout (cKO) were distinct from the regions affected by developmentally dynamic H3K27me3.

**Figure supplement 2.** Correlations of chromatin immunoprecipitation sequencing (ChIP-seq) signals across replicates.

datasets at P0 were less consistent than the E14 and P39 samples (*Figure 6—figure supplement 2*). As a result, our data were not well powered to detect changes at P0 and we chose to focus on the E14 vs. P39 DM regions as sites of developmental chromatin remodeling. Genes associated with these DM regions were enriched in biological processes involved in development such as nervous system

development and neurogenesis (*Figure 6—figure supplement 1B*). The developmental H3K27me3 DM regions overlapped developmental DMRs, with a notable overlap of regions gaining both mCG and H3K27me3 (*Figure 6B*). Both modifications may thus act together to repress thousands of genomic regions during development.

The P39 *Dnmt3a* cKO changes of H3K27me3 signal were weakly correlated with the developmental changes of H3K27me3 signal (r = –0.12, p<2.2e-16, *Figure 6—figure supplement 1C*). Moreover, there was little appreciable difference between the H3K27me3 signal fold-changes between P39 and E14 cKO samples compared with those of control samples (Spearman r=0.67, p<2.2e-16, *Figure 6—figure supplement 1D*).

To further stratify the joint distribution of developmental and *Dnmt3a* cKO-dependent changes in both H3K27me3 and DNA methylation, we assigned peaks to three groups (*Figure 6C–E* and *Figure 6—figure supplement 1E-G*). 'Group DevLoss' and 'Group DevGain' peaks lose or gain H3K27me3 during development, respectively (*Figure 6C–D* and *Figure 6—figure supplement 1E-F*). 'Group cKO' peaks have higher H3K27me3 in the *Dnmt3a* cKO compared to control at P39 (*Figure 6E* and *Figure 6—figure supplement 1G*). We found that developmental peaks (DevLoss and DevGain) were relatively unaffected by the cKO (ΔH3K27me3=0.02, –0.03 respectively, in units of $\log_{10}$(CPM +1)), whereas Group cKO had 4.5-fold larger mean effect (ΔH3K27me3=0.09). Group cKO peaks also experienced greater loss of mCG (ΔmCG = –23.5%) than Group DevLoss (–13.0%) or DevGain (18.6%) (*Figure 6C–E* and *Figure 6—figure supplement 1E-G*, middle and right panels). These results suggest that regions prone to alteration of H3K27me3 by *Dnmt3a* cKO are distinct from the regions affected by developmentally dynamic H3K27me3. We observed a very similar pattern when we examined sites with differential H3K27me3 between P0 and P39 (data not shown).

## Novel DNA methylation valleys with increased H3K27me3 signal in the *Dnmt3a* cKO

DNA methylation and H2K27me3 have complementary roles at DNA methylation valleys (DMVs), that is, large regions (≥5 kb) with low mCG (≤15%) that occur around key transcriptional regulators of development in human and mouse tissues (*Mo et al., 2015*; *Xie et al., 2013*). Previous studies comparing the epigenetic profile of DMVs across tissues identified multiple categories, including constitutive DMVs present in all tissues as well as tissue-specific DMVs (*Li et al., 2018*). We found more than twice as many DMVs in P39 cKO (1838) compared with control (881) neurons (*Supplementary file 10*), covering a greater genomic territory (16.02 Mbp in cKO, 7.91 Mbp in control). The majority of these P39 DMVs were either expanded or unique in cKO (*Figure 7A*), while DMVs identified in P0 samples were mostly not altered (*Figure 7—figure supplement 1*). Most P39 DMVs had active histone marks (H3K27ac+, H3K27me3-), while some had repressed or bivalent profiles consistent with PRC2-associated gene silencing (H3K27me3+) (*Figure 7B*).

By clustering the DMVs using their pattern of DNA methylation, chromatin modifications, and gene expression, we found eight distinct categories (*Figure 7C–D*). Whereas most DMVs lack H3K27me3 (clusters C2,3,5,7,8), we found some groups of DMVs associated with moderate (C1, C4) or high (C6) levels of H3K27me3. Cluster C6 DMVs, such as the promoter of *Tfap2c* (*Figure 7C–D*), had high H3K27me3 and low mCG in both control and cKO neurons, and were not strongly affected by the loss of mC in the *Dnmt3a* cKO. By contrast, cluster C1 and C4 DMVs, including the promoters of *Lhx2, Foxp1, Foxp2, Slc17a6* (encoding vesicular glutamate transporter, Vglut2) and *Sema3f*, gain mC during normal development (*Figure 7C–D*). Cluster C1 DMVs gained H3K27me3 in the P39 cKO compared to control animals. The loss of mC in these regions in the *Dnmt3a* cKO did not lead to strong activation of gene expression, potentially due to compensatory PRC2-mediated repression.

The remaining clusters lack H3K27me3 and are instead marked by low mCG and either high H3K27ac (cluster C8) or both H3K27ac and H3K4me3 (clusters C7). Cluster C5 DMVs (e.g. *Nr1d1, Pde4d*, *Figure 7D*) have high mCG at P0 and lose methylation in excitatory neurons during brain development. This demethylation is not affected in the *Dnmt3a* cKO, and we found little difference between the control and cKO neurons at these sites. Finally, clusters C2 and C3 were enriched for up- and down-regulated genes, respectively.

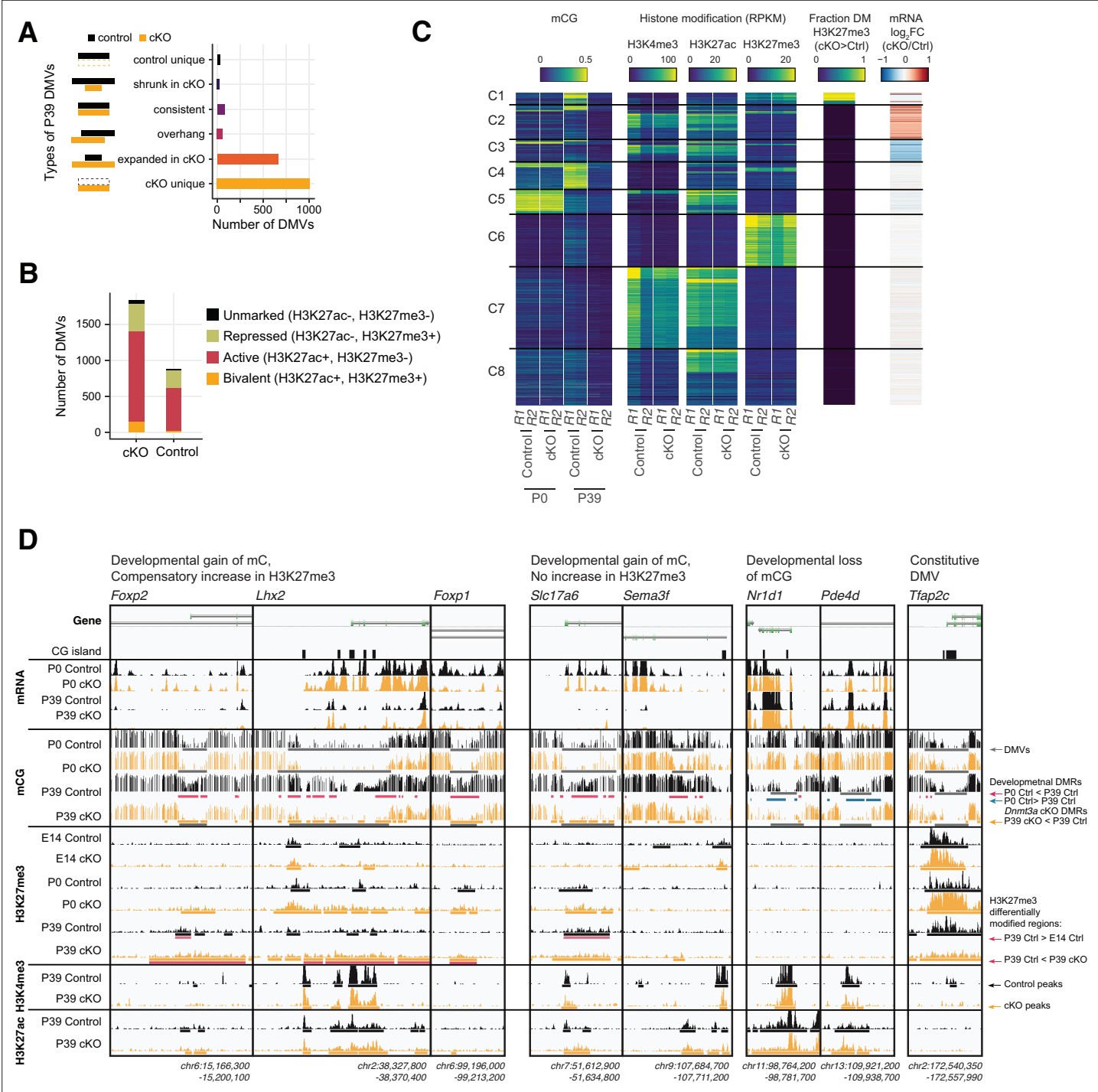

**Figure 7.** Distinct clusters of DNA methylation valleys (DMVs) were associated with the increased H3K27me3 signal in the *Dnmt3a* conditional knockout (cKO). (**A**) Number of DMVs identified in the P39 *Dnmt3a* cKO and the control samples, categorized by whether they appear in one or both groups or change size in the cKO. (**B**) Overlap of DMVs with the H3K4me4 and/or the H3K27me3 chromatin immunoprecipitation sequencing (ChIP-seq) peaks. (**C**) Heatmap of DMVs clustered by their methylation levels and histone modifications. The last two columns show the enrichments of differentially modified (DM) peaks of H3K27me3 and differentially expressed (DE) genes. R1/2, replicate 1/2; RPKM, reads per kilobase per million. (**D**) Browser tracks show examples of unique DMVs in the *Dnmt3a* cKO samples and the increased H3K27me3 signal in their flanking regions.

The online version of this article includes the following figure supplement(s) for figure 7:

**Figure supplement 1.** Number of DNA methylation valleys (DMVs) identified in the P0 *Dnmt3a* conditional knockout (cKO) and the control samples.

# Discussion

To examine the role of DNA methylation in cortical excitatory neurons after their birth, we developed a mouse model where the loss of DNA methylation occurs in postmitotic neurons, prior to the post-natal increase in *Dnmt3a* expression and non-CG methylation (*Lister et al., 2013*). *Neurod6* conditional deletion of *Dnmt3a* in cortical excitatory neurons abolished non-CG methylation and reduced CG methylation throughout the genome (*Figure 3C*). We found that *Dnmt3a* cKO neurons had altered expression of dozens of genes, including both up- and down-regulated genes (*Figure 3A–B*). Similar complex patterns in gene expression were reported when *Dnmt3a* was deleted in inhibitory neurons (*Lavery et al., 2020*) and following manipulation of the DNA methylation reader MeCP2 (*Boxer et al., 2020*; *Johnson et al., 2017*; *Lavery et al., 2020*). These gene expression changes may partly reflect the direct effect of lower gene body CH methylation and loss of CG and CH methylation at gene promoters and distal enhancers (*Boxer et al., 2020*; *Clemens et al., 2020*). In addition, some gene expression changes could result from the disruption of other regulatory processes, such as TF expression or chromatin modification. Indeed, we found that the *Dnmt3a* cKO DMRs overlapped with regions that gain methylation during normal postnatal development (*Figure 3G–H*). These DMRs had increased H3K27ac in the *Dnmt3a* cKO (*Figure 5C*), consistent with observations in adult animals lacking *Dnmt3a* specifically in GABAergic *Sst*- or *Vip*-expressing interneurons (*Stroud et al., 2020*). In those experiments, embryonic gene regulatory elements had lower cytosine methylation and increased H3K27ac and H3K4me1 (*Stroud et al., 2020*). This suggests an essential role for *Dnmt3a* and DNA methylation in shaping the transcriptome during development in part via the inactivation of embryonic enhancers, with potentially long-lasting effects on the gene expression pattern of mature neurons. Although we did not detect individual peaks with a statistically significant difference in H3K27ac between cKO and control (*Figure 4—figure supplement 1C*), this could be due to a small number of replicates and the subtle magnitude of changes in gene expression relative to the large number of tested peaks. Future experiments using more replicates or orthogonal techniques like CUT&RUN and CUT&Tag could help to further investigate the relationship between the *Dnmt3a*-dependent DNA methylation and the activation of enhancers.

There is a strong antagonistic relationship between DNA methylation and the PRC2-associated histone mark, H3K27me3 (*Brinkman et al., 2012*; *Jermann et al., 2014*; *Lynch et al., 2012*; *Reddington et al., 2013*; *Wu et al., 2010*). Switching between polycomb- and DNA methylation-mediated repression has been observed during development and in cancer (*Mohn et al., 2008*; *Schlesinger et al., 2007*; *Widschwendter et al., 2007*). Severe depletion of mCG can lead to redistribution of H3K27me3, causing derepression of developmental regulators such as the *Hox* gene clusters (*Reddington et al., 2013*). We did not observe ectopic expression of these genes, possibly due to the relatively modest reduction in mCG in our model compared with cells lacking *Dnmt1*. Instead, we found that in cortical excitatory neurons, thousands of sites gained H3K27me3 following the loss of mCG in the *Dnmt3a* cKO (*Figure 4—figure supplement 1D*). These sites, which normally gain DNA methylation during postnatal development, were left unmethylated in cKO neurons (*Figures 4B and 6E*). These regions were largely distinct from the sites that gain or lose H3K27me3 during normal development (*Figure 6C–E*), and had an intermediate level of H3K27me3 in the control samples (*Figure 4—figure supplement 3A*). A subset of these regions formed large-scale DMVs spanning key regulatory genes (*Figure 7C–D*). Overall, our results suggest that when DNA methylation is disrupted, H3K27me3 might partially compensate for the loss of mCG and/or mCH and act as an alternative mode of epigenetic repression. Nevertheless, we did not find differential expression in any of the four core components of PRC2 (*Ezh2*, *Suz12*, *Eed*, and *Rbbp4*) in adult *Dnmt3a* cKO animals. It is possible that the increased H3K27me3 was mediated by transient expression of PRC2 components during development in the cKO. Furthermore, the predictions from BART (*Figure 4A*) were derived from various cell lines and tissues from the ENCODE project (*Davis et al., 2018*; *ENCODE Project Consortium, 2012*), suggesting that the potential PRC2 binding at our DEGs may normally happen in systems other than the brain or pyramidal neurons, or at other time points during development. Additional experiments which directly manipulate components of the PRC2 system are required to further test the potential compensation mechanism.

Previous work showed that the early embryonic deletion of *Dnmt3a* in excitatory neurons caused gross motor deficits and a shortened lifespan (*Nguyen et al., 2007*). Mid-gestation *Neurod6*-driven *Dnmt3a* ablation, however, did not cause such alterations, allowing us to test how the epigenetic

and transcriptional alterations affected the morphology and function of excitatory synapses, neurons' passive and active properties, and behavior. We found that prelimbic layer 2 neurons of *Dnmt3a* cKO animals had more immature dendritic spines and were less sensitive to somatic injections of depolarizing current compared to control neurons (*Figure 2*). This is consistent with several of the observed DE genes having annotated roles in dendrite morphogenesis (*Elavl4*, *Hecw2*, *Ptprd*, *Figure 3B*) and Na⁺ influx/transport (*Hecw2*, *Scn3b*, *Figure 3B*). Indeed, the down-regulation of the latter set of genes could be expected to increase the action potential threshold (thus dampening neuronal excitability) and trigger neurodevelopmental delays (*Berko et al., 2017*). As well, *Cacnb3*, a gene that encodes a regulatory beta subunit of the voltage-dependent calcium channel, was found among the set of down-regulated genes after loss of *Dnmt3a*. This gene has been linked with schizophrenia (*Maycox et al., 2009*), ADHD, and bipolar disorder (*van Hulzen et al., 2017*) in humans. Our study thus suggests that the disruption of methylation patterns established by *Dnmt3a* during infancy might have far-reaching mechanistic relevance for multiple neurodevelopmental disorders, complementing previous genetic risk association studies linking *Dnmt3a* with autism (*C Yuen et al., 2017*; *Sanders et al., 2015*). Although our experiments were not designed to make quantitative comparisons between cohorts of different sexes, we observed a wider range of behavioral impairments in male *Dnmt3a* cKO mice, which included, for example, decreased social interest (*Figure 1*). Those data correlate well with several human neurodevelopmental disorders, in which males are disproportionately more affected than females, and thus provide important support for follow-up investigations into the underlying causes of those differences.

Our findings highlight the critical and interconnected roles in brain development and cognitive function of two major modes of epigenetic repression of gene expression: DNA methylation and PRC2-mediated repression. The loss of DNA methylation in excitatory neurons has effects on gene expression, synaptic function, and cognitive behavior. Moreover, loss of DNA methylation leads to a gain of the PRC2-associated repressive mark H3K27me3. Our cKO is a restricted manipulation of one neuron type, yet it directly impacts DNA methylation throughout the genome at millions of sites. PRC2-mediated repression may compensate for the loss of mCG and/or mCH, acting as an alternative repressive mechanism when DNA methylation is disrupted. Future work focusing on earlier developmental stages, and using targeted methods to manipulate epigenetic marks in local genomic regions (*Liu et al., 2016*), may help elucidate the causal interactions among epigenetic modifications that are critical for neuronal maturation and function.

## Materials and methods

### Generation of the *Dnmt3a* cKO mice line

All animal procedures were conducted in accordance with the guidelines of the American Association for the Accreditation of Laboratory Animal Care and were approved by the Salk Institute for Biological Studies Institutional Animal Care and Use Committee (protocol number 18-00006). For behavior, slice physiology, and spine analyses, *Dnmt3a*-floxed animals (*Okano et al., 1999*) (backcrossed to C57BL/6 for at least seven generations) were crossed to *Neurod6*-Cre (*Nex*-Cre) (*Goebbels et al., 2006*, backcrossed to C57BL/6 J for >10 generations) mice to generate *Dnmt3a*-KO animals carrying the deletion only in pyramidal cells. To be able to isolate pyramidal neuron nuclei for DNA methylation, transcription, and ChIP analyses, the mouse lines (*Dnmt3a*-KO and *Neurod6*-Cre) were crossed to a mouse line carrying the INTACT background (B6.129-*Gt(ROSA)26Sor*^tm5(CAG-Sun1/sfGFP)Nat/MmbeJ^.Strain 030952, Jackson laboratories). The deletion of *Dnmt3a* from pyramidal cells was confirmed by RNA-seq (deletion of exon 19) and Western blot (*Figure 1B* and *Figure 1—figure supplement 3A-B*). For both backgrounds, Nex-Cre hemizygous mice were used as controls.

### Frontal cortex dissection, nuclei isolation, and flow cytometry

Frontal cortex tissue was produced as described (*Lister et al., 2013*; *Luo et al., 2017*) from postnatal day 0 and 39 (P39) *Dnmt3a* cKO and control animals, in an INTACT background. The nuclei of GFP-expressing NeuN-positive excitatory neurons were isolated and collected using FANS as described (*Lister et al., 2013*; *Luo et al., 2017*) with the following modification: prior to FANS, nuclei were labeled with anti-NeuN-AlexaFluor647 and anti-GFP-AlexaFluor488. Nuclei were sorted as described

(*Lister et al., 2013*). Double positive nuclei were retained for RNA-seq, ChIP-seq, and MethylC-seq library preparation and sequencing.

## Western blot

Frontal cortex proteins were obtained by homogenization in RIPA buffer of the following composition: 150 mM NaCl, 10 mM $Na_2HPO_4$, 1% NaDOC, 1% NP-40, 0.5% SDS, 1 mM DTT, 1 mM PMSF in DMSO, supplemented with protease inhibitor (Sigma-Aldrich #11836153001) and phosphatase inhibitor (Pierce #A32957) cocktails. After centrifugation at 15,000× *g*, supernatants were preserved and protein concentration was determined by the BCA method (Pierce). Protein bands were separated in 8% PAGE gels and transferred to nitrocellulose membranes. After blocking in TBS-tween with 5% milk, DNMT3A was detected by the use of anti-DNMT3A antibody (Abcam) and chemiluminescence. DNMT3A bands were normalized to ACTIN content in each sample.

## Patch-clamp electrophysiology

Male and female mice (6–9 weeks) were anesthetized with isoflurane and decapitated. The brains were quickly removed and coronal slices of the frontal cortex containing the prelimbic region (~2 mm anterior to Bregma) were cut in an ice-cold slicing medium of the following composition (in mM): 110 sucrose, 2.5 KCl, 0.5 $CaCl_2$, 7 $MgCl_2$, 25 $NaHCO_3$, 1.25 $NaH_2PO_4$, and 10 glucose (bubbled with 95% $O_2$ and 5% $CO_2$). The slices were then transferred to artificial CSF (aCSF) containing (in mM): 130 NaCl, 2.5 KCl, 1.25 $NaH_2PO_4$, 23 $NaHCO_3$, 1.3 $MgCl_2$, 2 $CaCl_2$, and 10 glucose, equilibrated with 95% $O_2$ and 5% $CO_2$ at 35°C for 30 min and afterward maintained at room temperature (22–24°C) for at least 1 hr (patch-clamp recording) before use. Brain slices were then transferred to a recording chamber and kept minimally submerged under continuous superfusion with aCSF at a flow rate of ~2 ml/min. Whole-cell recordings were obtained from putative prelimbic layer 2 (L2) pyramidal cells (identified by their pyramidal-shaped cell bodies and long apical dendrite using an upright microscope equipped with differential interference contrast optics). In acute mPFC slices, the prelimbic L2 is clearly distinguishable from L1 and L3 as a thin dark band that is densely packed with neuron somata. Pipettes had a tip resistance of 4–8 MΩ when filled with an internal solution of the following composition (in mM): 125 K-gluconate, 15 KCl, 8 NaCl, 10 HEPES, 2 EGTA, 10 $Na_2$ phosphocreatine, 4 MgATP, 0.3 NaGTP (pH 7.25 adjusted with KOH, 290–300 mOsm). Access resistance (typically 15–35 MΩ) was monitored throughout the experiment to ensure stable recordings.

After obtaining the whole-cell configuration in voltage-clamp mode, cells were switched from a holding potential of –70 mV to current-clamp mode and the bridge-balance adjustment was performed. Passive electrical properties were quantified from recordings with hyperpolarizing current injections that evoked small ~5 mV deflections in membrane potential from resting. Responses to stepwise current injections (10–300 pA in increments of 10 pA; duration, 1 s) were recorded at 20 kHz in order to calculate input-output curves and rheobase – the minimal current necessary to trigger the first action potential. Miniature excitatory postsynaptic currents (mEPSCs) were recorded for 5 min in voltage-clamp mode (Vh=−70 mV) in the presence of the Na+ channel blocker, TTX (0.5 µM), to prevent the generation of action potentials, and picrotoxin (50 µM), an antagonist of $GABA_A$ receptors, to minimize inhibitory responses. In these conditions, mEPSCs could be blocked by the AMPA receptor antagonist, CNQX (25 µM). Single events larger than 6 pA were detected offline using the Minianalysis program (Synaptosoft Inc Decatur, GA). All data were acquired using a Multiclamp 700B amplifier and pCLAMP 9 software (Molecular Devices, LLC, San Jose, CA).

## Fluorescent labeling of dendritic spines

Coronal brain slices containing the mPFC of 10–13 weeks' female mice were prepared as for electrophysiological recordings and placed in a beaker for 3 hr at room temperature (24°C) to allow functional and morphological recovery. One slice was then transferred to a recording chamber and kept minimally submerged under continuous superfusion with aCSF bubbled with carbogen (95% $O_2$/5% $CO_2$) at a flow rate of ~2 ml/min. Previously sonicated crystals of the fluorescent marker DiI were placed next to the somata of layer 2 neurons in the prelimbic cortex, identified with the aid of an upright microscope equipped with differential interference contrast optics. The mice in these experiments had a Thy1-YFP background to help rule out non-specific labeling of deeper layer neurons. The neurons were exposed to the DiI crystals for 60 min. The slices were then gently removed from the

incubation chamber with a transfer pipette and immersed in fixative (4% PFA) for 30 min. Then, the slices were rinsed three times with PBS for 5–10 min each, after which they were mounted on slides with prolonged gold antifade mounting medium (Life Technologies – Molecular Probes). The slides were kept in a dark box for 24 hr at room temperature to allow the liquid medium to form a semi-rigid gel. Imaging took place 24–48 hr from the time of the initial staining.

## Confocal imaging

Dendritic spines were imaged by an investigator blind to the genotype using a Zeiss AiryScan confocal laser scanning microscope. All images were taken using the Zeiss Plan-APOCHROMAT 63× oil-immersion lens (N/A 1.4). A 543 nm laser was used to visualize the fluorescence emitted by DiI. Serial stack images with a 0.2 µm step size were collected, and then projected to reconstruct a three-dimensional image that was post-processed by the AiryScan software. Dendritic segments in layer 1, which were derived from layer 2 pyramidal neurons retrogradely labeled with DiI and that were well separated from neighboring neural processes, were randomly sampled and imaged. Each dendritic segment imaged for quantification belonged to a different neuron.

## Dendritic spine quantification

The z-stack series were imported into the Reconstruct software (https://synapseweb.clm.utexas.edu/software-0/), with which a second investigator also blind to the genotype performed the identification of dendritic spines and their morphometric analysis. By scrolling through the stack of different optical sections, individual spine heads could be identified with greater certainty. All dendritic protrusions with a clearly recognizable stalk were counted as spines. Spine density was determined by summing the total number of spines per dendritic segment length (30–40 µm) and then calculating the average number of spines per µm. Individual dendritic spines were classified in the following order according to pre-established criteria: protrusion longer than 3 µm, filopodia; head wider than 0.6 µm, mushroom; protrusion longer than 2 µm and head narrower than 0.6 µm, long-thin; protrusion longer than 1 µm and head narrower than 0.6 µm, long-thin; the remaining spines were labeled stubby. Branched spines (with more than one neck) were counted separately.

## Behavioral testing

Phenotypic characterization was initiated when the animals reached 9 weeks of age using cohorts of 10–15 male or female mice per genotype, according to the order described below.

## Open field test

The open field test was performed using MED Associates hardware and the Activity Monitor software according to the manufacturer's instructions (MED Associates Inc, St Albans, VT). Animals were individually placed into clear Plexiglas boxes ($43.38 \times 43.38 \times 30.28$ cm³) surrounded by multiple bands of photo beams and optical sensors that measure horizontal and vertical activity. Movement was detected as breaks within the beam matrices and automatically recorded for 60 min.

## Light/dark transfer test

The light/dark transfer procedure was used to assess anxiety-like behavior in mice by capitalizing on the conflict between exploration of a novel environment and the avoidance of a brightly lit open field (150–200 lux in our experiments). The apparatus were Plexiglas boxes as for the open field test ($43.38 \times 43.38 \times 30.28$ cm³) containing dark box inserts ($43.38 \times 12.8 \times 30.28$ cm³). The compartments were connected by an opening ($5.00 \times 5.00$ cm²) located at floor level in the center of the partition. The time spent in the light compartment was used as a predictor of anxiety-like behavior, that is, a greater amount of time in the light compartment was indicative of decreased anxiety-like behavior. Mice were placed in the dark compartment (4–7 lux) at the beginning of the 15 min test.

## Elevated plus maze

The maze consisted of four arms (two open without walls and two with enclosed walls) 30 cm long and 5 cm wide in the shape of a plus sign. The apparatus was elevated approximately 33 cm over a table. At the beginning of each trial, one animal was placed inside a cylinder located at the center of the maze for 1 min. The mouse was then allowed to explore the maze for 5 min. The session was

video-recorded by an overhead camera and subjected to automated analysis using ANY-maze software. The apparatus was wiped down with sani-wipes between trials to remove traces of odor cues. The percentage of time spent in open or closed arms was scored and used for analysis.

## Y-maze test for spontaneous alternations

Spontaneous alternations between three 38 cm long arms of a Y-maze were taken as a measure of working memory. Single 6 min trials were initiated by placing each mouse in the center of the Y-maze. Arm entries were recorded with a video camera and the total number of arm entries, as well as the order of entries, was determined. The apparatus was wiped down with sani-wipes between trials to remove traces of odor cues. Spontaneous alternations were defined as consecutive triplets of different arm choices and % spontaneous alternation was defined as the number of spontaneous alternations divided by the total number of arm entries minus 2.

## Social approach

The apparatus consisted of a Plexiglas box ($60 \times 38 \times 23.5$ cm$^3$) divided into three compartments by Plexiglas partitions containing openings through which the mice could freely enter the three chambers. The test was conducted in two 10 min phases. In phase I, the test mouse is first allowed to explore the chambers for 10 min. Each of the two outer chambers contained an empty, inverted stainless steel wire cup. In phase II, the test mouse is briefly removed, and a sex-matched unfamiliar mouse was placed under one of the wire cups, and plastic blocks were placed under the other wire cup. The test mouse was then gently placed back in the arena and given an additional 10 min to explore. An overhead camera and video tracking software (ANY-maze, Wood Dale, IL) were used to record the amount of time spent in each chamber. The location (left or right) of the novel object and novel mouse alternates across subjects.

## Acoustic startle responses and PPI of the acoustic startle response

Acoustic startle responses were tested inside SR-LAB startle apparatus (San Diego Instruments, San Diego, CA), consisting of an inner chamber with a speaker mounted to the wall and a cylinder mounted on a piezoelectric sensing platform on the floor. At the beginning of testing, mice were placed inside the cylinder and then were subjected to background 65 dB white noise during a 5 min acclimation period. The PPI session began with the presentation of six pulse-alone trials of 120 dB, 40 ms. Then, a series of pulse-alone trials and prepulse trials (69, 73, or 81 dB, 20 ms followed by 100 ms pulse trial, 120 dB) were each presented 12 times in a pseudorandom order. The session concluded with the presentation of six pulse-alone trials. The apparatus was wiped down with sani-wipes between trials to remove traces of odor cues. The startle amplitude was calculated using arbitrary units, and the acoustic startle response was the average startle amplitude of pulse-alone trials. The percent PPI was calculated as follows: [100 − (mean prepulse response/mean pulse alone response) × 100].

## Cued and contextual fear conditioning

Fear conditioning experiments were performed using automated fear conditioning chambers (San Diego Instruments, San Diego, CA), similar to previous studies (*Gresack et al., 2010*; *Risbrough et al., 2014*). On day 1, after a 2 min acclimation period, mice were presented with a tone conditioned stimulus (75 dB, 4 kHz) for 20 s that co-terminated with a foot shock unconditioned stimulus (1 s, 0.5 mA). A total of three tone-shock pairings were presented with an inter-trial interval of 40 s. To assess acquisition, freezing was quantified during foot shock presentations. Mice were returned to their home cages 2 min after the final shock. These moderate shock parameters were previously found suitable to detect both increases and decreases in fear-conditioned behavior (*Risbrough et al., 2014*). Twenty-four hr later, on day 2, mice were re-exposed to the conditioning chamber to assess context-dependent fear retention. This test lasted 8 min during which time no shocks or tones were presented and freezing was scored for the duration of the session. Time freezing was quantified across four 2 min blocks. Day 3: 24 hr after the context fear-retention test, mice were tested for CS-induced fear retention and extinction. The context of the chambers was altered across several dimensions (tactile, odor, visual) for this test in order to minimize generalization from the conditioning context. After a 2 min acclimation period, during which time no tones were presented ('pre-tone'), 32 tones were presented for 20 s with an inter-trial interval of 5 s. Freezing was scored during each tone presentation and

quantifications were done in eight blocks of four tones. Mice were returned to their home cage immediately after termination of the last tone. On day 4, after a 2 min acclimation period, during which time no tones were presented ('pre-tone'), a shorter session of 16 tones was used to assess extinction. Time freezing was quantified across four blocks of four tones.

## RNA extraction, RNA-seq library construction, and sequencing

Nuclei (between 50,000 and 60,000) were used to isolate RNA using Single-Cell RNA Purification Kit (Norgen, catalog# 51800). In brief, aliquots of nuclei were resuspended in 350 µl of RL buffer (Norgen) and passed through an 18 G syringe five times. RNA extraction, including DNase digestion, followed manufacturers' instructions. RNA was eluted in 20 µl of Elution Solution A (Norgen). The nuclear RNA concentration was determined using TapeStation (Agilent). RNA was diluted to 1 ng/µl and a total of 5 ng was processed for RNA-seq library preparation. RNA libraries were prepared using NuGen Ovation RNA-Seq System V2 (#7102–32) for cDNA preparation following the product manual. cDNA purification was done using Zymo Research DNA Clean & Concentrator-25 with modification from the Ovation protocol. cDNA, eluted in 30–40 µl of TE (1 µg per sample), was fragmented at 300 bp using Covaris S2 (Sonolab S-series V2), followed by library preparation according to KAPA LTP Library Preparation Kit (KK8232), using Illumina indexed adapters. Libraries were sequenced on NovaSeq 6000.

## DNA extraction

DNA extraction was performed using the Qiagen DNeasy Blood and Tissue kit (catalog #69504) and eluted into 50–100 µl AE.

## Genomic DNA library construction and sequencing

1.5 µg of genomic DNA was fragmented with a Covaris S2 (Covaris, Woburn, MA) to 400 bp, followed by end repair and addition of a 3' A base. Cytosine-methylated adapters provided by Illumina (Illumina, San Diego, CA) were ligated to the sonicated DNA at 16°C for 16 hr with T4 DNA ligase (New England Biolabs). Adapter-ligated DNA was isolated by two rounds of purification with AMPure XP beads (Beckman Coulter Genomics, Danvers, MA). Half of the adapter-ligated DNA molecules were enriched by 6 cycles of PCR with the following reaction composition: 25 µl of Kapa HiFi Hotstart Readymix (Kapa Biosystems, Woburn, MA) and 5 µl TruSeq PCR Primer Mix (Illumina) (50 µl final). The thermocycling parameters were: 95°C 2 min, 98°C 30 s, then 6 cycles of 98°C 15 s, 60°C 30 s, and 72°C 4 min, ending with one 72°C 10 min step. The reaction products were purified using AMPure XP beads and size selection was done from 400 to 600 bp. Libraries were sequenced on NovaSeq 6000.

## MethylC-seq library construction and sequencing

MethylC-seq libraries were prepared as previously described (*Urich et al., 2015*). All DNA obtained from the extraction was spiked with 0.5% unmethylated Lambda DNA. The DNA was fragmented with a Covaris S2 (Covaris, Woburn, MA) to 300 bp, followed by end repair and addition of a 3' A base. Cytosine-methylated adapters provided by Illumina (San Diego, CA) were ligated to the sonicated DNA at 16°C for 16 hr with T4 DNA ligase (New England Biolabs). Adapter-ligated DNA was isolated by two rounds of purification with AMPure XP beads (Beckman Coulter Genomics, Danvers, MA). Adapter-ligated DNA (≤450 ng) was subjected to sodium bisulfite conversion using the EZ methylation Direct kit (Zymo, D5021) as per the manufacturer's instructions. The bisulfite-converted, adapter-ligated DNA molecules were enriched by 8 cycles of PCR with the following reaction composition: 25 µl of Kapa HiFi Hotstart Uracil +Readymix (Kapa Biosystems, Woburn, MA) and 5 µl TruSeq PCR Primer Mix (Illumina) (50 µl final). The thermocycling parameters were: 95°C 2 min, 98°C 30 s, then 8 cycles of 98°C 15 s, 60°C 30 s and 72°C 4 min, ending with one 72°C 10 min step. The reaction products were purified using AMPure XP beads. Up to two separate PCR reactions were performed on subsets of the adapter-ligated, bisulfite-converted DNA, yielding up to two independent libraries from the same biological sample. MethylC-seq libraries were sequenced on NovaSeq 6000.

## ChIP-seq library construction and sequencing

Sorted nuclei were crosslinked for 15 min in 1% formaldehyde solution and quenched afterward with glycine at a final concentration of 0.125 M. After crosslinking, nuclei were sonicated in Lysis buffer

(50 mM Tris HCl pH 8, 20 mM EDTA, 1% SDS, 1× EDTAfree protease inhibitor cocktail). ChIP assays were conducted with antibodies against H3K27me3 (39156, Active Motif), H3K27ac (39133, Active Motif), and H3K4me3 (04-745, Millipore Sigma). Mouse IgG (015-000-003, Jackson ImmunoResearch) served as a negative control. H3K4me3 ChIP-seq assays were conducted with 100 K nuclei and 500 K nuclei were used for H3K27me3 and H3K27ac ChIP-seq assays. The respective antibodies and IgG were coupled for 4–6 hr to Protein G Dynabeads (50 µl, 10004D, Thermo Fisher Scientific). Equal amounts of sonicated chromatin were diluted with 9 volumes of Binding buffer (1% Triton X-100, 0.1% Sodium Deoxycholate, 2× EDTA free protease inhibitor cocktail) and subsequently incubated overnight with the respective antibody-coupled Protein G beads. Beads were washed successively with low salt buffer (50 mM Tris HCl pH 7.4, 150 mM NaCl, 2 mM EDTA, 0.5% Triton X-100), high salt buffer (50 mM Tris HCl pH 7.4, 500 mM NaCl, 2 mM EDTA, 0.5% Triton X-100) and wash buffer (50 mM Tris HCl pH 7.4, 50 mM NaCl, 2 mM EDTA) before de-crosslinking, proteinase K digestion, and DNA precipitation. Libraries were generated with the Accel-NGS 2S Plus DNA Library Kit (21024, Swift Biosciences) and sequenced on the Illumina HiSeq 4000 Sequencing system.

## RNA-seq data processing

RNA-seq reads first went through quality control using FastQC (**Andrews et al., 2012**) (v0.11.8, https://www.bioinformatics.babraham.ac.uk/projects/fastqc/), and then were trimmed to remove sequencing adapters and low-quality sequences (minimum Phred score 20) using Trim Galore (v0.5.0, https://www.bioinformatics.babraham.ac.uk/projects/trim_galore/, a wrapper tool powered by Cutadapt [**Martin, 2011**] v1.16) in the paired-end mode. Clean reads were then mapped to the mouse mm10 (GRCm38) genome and the GENCODE annotated transcriptome (release M10) with STAR (**Dobin et al., 2013**) (Spliced Transcripts Alignment to a Reference, v2.5.1b). Gene expression was estimated using RSEM (**Li and Dewey, 2011**) (RNA-Seq by Expectation Maximization, v1.2.30). Gene-level 'expected count' from the RSEM results were rounded and fed into edgeR (**Robinson et al., 2010**) (v3.24.1) to call DE genes. Only genes that were expressed (with counts per million [CPM] > 2) in at least two samples were kept. These counts were then normalized using the TMM method (**Robinson and Oshlack, 2010**), and DE genes were then called in the quasi-likelihood F-test mode, requiring FDR < 0.05 and FC > 20% (**Supplementary file 2**). To quantify TE expression, we used the Fetch, Clean, Map, and Count modules from SQuIRE (**Yang et al., 2019**) with the RepeatMasker annotation from UCSC. To evaluate the differences in TE family and class abundance between *Dnmt3a* knockout and control samples, we grouped SQuIRE's TE subfamily expression estimates (FPKM) into their respective families and classes and performed two-sample t-tests.

## Enrichment test of GO terms in DE genes

Gene Ontology (GO) enrichment analysis was performed using clusterProfiler (**Yu et al., 2012**) (v3.10.0). Only 'Biological Process' terms with no less than 10 genes and no more than 250 genes were considered. Terms with FDR < 0.05 were considered significantly enriched.

## Genomic DNA sequencing data processing and SNP calling

To estimate the completeness of the inbreeding of the mouse strains, and to avoid incorrect cytosine context assignment in the following MethylC-seq data processing, we used the genomic DNA sequencing data of both the *Dnmt3a* cKO and the control animals to call SNPs against the mouse mm10 genome. We followed the GATK 'best practices for germline SNPs and indels in whole genomes and exomes' pipeline (**DePristo et al., 2011**; **McKenna et al., 2010**; **Van der Auwera et al., 2013**). Briefly, raw data were first trimmed to remove sequencing adapters and low-quality sequences (minimum Phred score 20) using Trim Galore in the paired-end mode. Clean data were then mapped to the mm10 genome using BWA (**Li and Durbin, 2009**) (v0.7.13-r1126). Duplicates reads were marked with Picard (**Broad Institute, 2018**). Then the analysis-ready reads were fed into GATK (v3.7) to perform two rounds of joint genotyping and base recalibration. Variants were then filtered using the following criteria: QD < 2.0 || FS > 60.0 || MQ < 40.0 || MQRankSum < –12.5 || ReadPosRankSum < –8.0 || SOR > 4.0. By that, we identified 548,530 and 507,669 SNPs (relative to mm10) in the *Dnmt3a* cKO and the control animals, respectively. At last, we created a substituted genome to mask out all these SNPs (replaced with Ns) with the 'maskfasta' tool in the BEDTools suite (**Quinlan and Hall, 2010**) (v2.27.1). This substituted genome was used in the following MethylC-seq data processing pipeline.

## MethylC-seq data processing

MethylC-seq reads were processed using the methylpy pipeline (v1.3.2, https://github.com/yupenghe/methylpy; *He, 2021*) as previously described (*Lister et al., 2013*; *Mo et al., 2015*). Briefly, a computationally bisulfite-converted genome index was built using the aforementioned substituted genome file appended with the lambda phage genomic sequence. MethylC-seq raw reads were first trimmed to remove sequencing adapters and low-quality sequences (minimum Phred score 10) using Cutadapt in paired-end mode. To acquire higher mappability, we treated the two ends of the clean reads as they were sequenced in single-end mode, and mapped them to the converted genome index with bowtie2 (*Langmead and Salzberg, 2012*) (v2.3.0) as aligner in the single-end pipeline of methylpy. Only reads uniquely mapped were kept, and clonal reads were removed. The bisulfite non-conversion rate (NCR) was estimated using the spiked-in unmethylated lambda phage DNA. For each cytosine, a binomial test was performed to test whether the methylation levels are significantly greater than 0 with an FDR threshold of 0.01.

For a particular genomic region, the raw methylation level for a given cytosine context (CG or CH) was defined as:

$$\%mC \; = \; 100 \times \frac{m}{h},$$

where m is the total number of methylated based calls within the region, and h is the total number of covered based calls within the region. Methylation levels were then corrected for NCR using the following maximum likelihood formula:

$$\%mC\_adj = \frac{\%mc - \%NCR}{100 - \%NCR}, \text{ where } \%mC\_adj \in \left[0, \; 100\right].$$

When profiling the methylation landscapes around DE genes, we selected control genes with comparable gene expression using the R package MatchIt (*Ho et al., 2011*) (v3.0.2). These control genes were defined with nearest neighbor matching of the expression (in the unit of TPM) using logistic link propensity score as a distance measure, requiring the standard deviation of the distance to be less than 0.01.

## DMRs calling

CG DMRs were identified using a previously reported method (*Ma et al., 2014*; *Schmitz et al., 2013*; *Schultz et al., 2015*), which is implemented in the DMRfind function in methylpy. We required at least three differentially methylated sites within a particular DMR, and significant sites located within 500 bp of each other were merged into the same DMR. With an FDR cutoff of 0.01 and a post-filtering cutoff of methylation levels change greater than 30 (in the unit of %mCG), we found 222,006 *Dnmt3a* cKO hypo-DMRs in P39 (*Supplementary file 3*). Note that with these criteria we also found 89 *Dnmt3a* cKO hyper-DMRs, which we thought were noise and/or SNPs that failed to be detected by the masking pipeline described earlier. Therefore, we removed these hyper-DMRs from further consideration.

## Enrichment test of DMRs and other genomic regions

To test whether DMRs were significantly enriched in certain genomic features, we use methods adapted from a recent report (*Rizzardi et al., 2019*). Briefly, for each genomic feature, we constructed a 2 by 2 contingency table of ($n_{11}$, $n_{12}$, $n_{21}$, $n_{22}$), where:

- $n_{11}$ is the number of CG sites in DMRs that were inside the feature;
- $n_{12}$ is the number of CG sites in DMRs that were outside of the feature;
- $n_{21}$ is the number of CG sites not in DMRs that were inside in feature;
- $n_{22}$ is the number of CG sites not in DMRs that were outside of the feature.

The total number of CG sites in consideration was the number of autosomal and chromosome X CG in the reference genome. Counting the number of CG rather than the number of DMRs or bases accounts for the non-uniform distribution of CG along the genome and avoids double-counting DMRs that are both inside and outside of the feature.

With this contingency table, we estimated the enrichment log odd ratio (OR) along with its standard error (se) and 95% confidence interval (ci) with the following formulas:

$$\log_2 \text{OR} = \log_2 n_{11} + \log_2 n_{12} - \log_2 n_{21} - \log_2 n_{22}$$

$$\text{se}\left(\log_2 \text{OR}\right) = \sqrt{1/n_{11} + 1/n_{12} + 1/n_{21} + 1/n_{22}}$$

$$\text{ci}\left(\log_2 \text{OR}\right) = \left[\log_2 \text{OR} - 2 \times \text{se}\left(\log_2 \text{OR}\right), \; \log_2 \text{OR} + 2 \times \text{se}\left(\log_2 \text{OR}\right)\right]$$

p-Value from performing Fisher's exact test for testing the null of independence of rows and columns in the contingency table (the null of no enrichment or depletion) was computed using the fisher.test() function in R.

The genomic regions/features used in these enrichment tests include: a list of developmental DMRs that gain or lose methylation during development (*Lister et al., 2013*); gene features (genic, exonic, intronic, promoter, 5'UTR, 3'UTR, intergenic) based on the GENCODE vM10 annotation (promoters were defined as the ±2 kb regions around transcription start sites); CpG island (CGI)-related features based on the 'cpgIslandExt' annotation from the UCSC genome browser (*Karolchik et al., 2004*; *Kent et al., 2002*) (http://genome.ucsc.edu/index.html), where CGI shores were defined as CGI ± 2 kb, CGI shelves were defined as ±2–4 kb of CGI and open seas were defined as regions that were at least 4 kb away from any CGI; the 12 states of the chromatin states map in mouse embryonic stem cell (*Pintacuda et al., 2017*) (https://github.com/guifengwei/ChromHMM_mESC_mm10) generated by ChromHMM (*Ernst and Kellis, 2017*; *Ernst and Kellis, 2012*) using ChIP-seq data from the ENCODE project (*Davis et al., 2018*; *ENCODE Project Consortium, 2012*); the H3K4me3, H3K27ac, and H3K27me3 peaks and the H3K27me3 differentially binding regions generated with our ChIP-seq data (see the 'ChIP-seq data processing' section).

## Predicting functional TFs regulating the DE genes with BART

The lists of up-regulated and down-regulated genes were fed into BART (*Wang et al., 2016*; *Wang et al., 2018*) (BART) separately to predict functional TFs and chromatin regulators that bind at cis-regulatory regions of the DE genes. To make a better visualization, we transformed the relative rank (a metric generated by BART to represent the average rank of Wilcoxon p-value, z-score, and max AUC for each factor divided by the total number of factors) into the functional TF rank score (which is simply 1 minus the relative rank) so that the higher the rank score the more possible the TF regulates the DE genes. The integrative rank significance was estimated with the Irwin-Hall p-value. TFs with Irwin-Hall p-value < 0.05 were considered significant.

## Enrichment test of known TF binding motifs in DMRs

We used the 'findMotifsGenome.pl' tool in HOMER (*Heinz et al., 2010*) (Hypergeometric Optimization of Motif EnRichment, v4.8.3) to find known TF binding motif in the *Dnmt3a* cKO DMRs. The parameters used are as follows: '-size 500 -len 8,10,12S 25 -fdr 100p 10 -mset vertebrates -bits -gc -nlen 3 -nomotif'. A set of non-neural DMRs (*Hon et al., 2013*) was used as the background.

## ChIP-seq data processing

ChIP-seq reads were pre-processed with the ENCODE Transcription Factor and Histone ChIP-Seq processing pipeline (https://github.com/ENCODE-DCC/chip-seq-pipeline2, v1.1.6; *Jin wook, 2022*). Briefly, paired-end reads were mapped to the mm10 genome with BWA (*Li and Durbin, 2009*) (v0.7.13-r1126). Reads were then filtered using samtools (*Li et al., 2009*) (v1.2) to remove unmapped, mate unmapped, not primary alignment and duplicate reads (-F 1804). Properly paired reads were retained (-f 2). Multi-mapped reads (MAPQ < 30) were removed. PCR duplicates were removed using the MarkDuplicates tool in Picard (*Broad Institute, 2018*) (v2.10.6). Reads mapped to the blacklist regions (*ENCODE Project Consortium, 2012*) in the mouse mm10 genome (http://mitra.stanford.edu/kundaje/akundaje/release/blacklists/mm10-mouse/mm10.blacklist.bed.gz) were also removed.

Peak calling was performed using epic2 (*Stovner and Sætrom, 2019*) (v0.0.16), a reimplementation of SICER (*Zang et al., 2009*). For H3K4me3 and H3K27ac, we used the following parameters: '--bin-size 200 --gaps-allowed 1'. For H3K27me3, we used the following parameters: '--bin-size 200 --gaps-allowed 3'. The IgG sample was used as a control.

DM regions of the histone modification ChIP-seq data were called using DiffBind (*Ross-Innes et al., 2012*) (v2.10.0) in DESeq2 (*Love et al., 2014*) (v1.22.1) mode. Regions with FDR < 0.05 were considered significant. Genes associated with these regions were identified using GREAT (*McLean et al., 2010*) (v3.0.0) with the 'Basal plus extension' association rule with default parameters. GO

enrichment analysis of the associated genes was performed with GREAT, and we considered the GO biological process terms with hypergeometric test FDR < 0.05 as significant.

To select a set of non-DM control regions to match the base levels of H3K27me3 in DM regions, we started with the union peaks of the control and cKO samples and removed any peaks that overlap the DM regions. From these non-DM peaks, we used the R package MatchIt (*Ho et al., 2011*) (v3.0.2) to select regions with matched peak lengths and H3K27me3 levels (in the unit of RPKM) as those in the DM regions, with the greedy nearest neighbor matching using logistic link propensity score as a distance measure (requiring the standard deviation of the distance to be less than 0.01).

## Definition of bivalent and active CGI promoters

CGI promoters were defined as CpG islands (downloaded from the UCSC genome browser) that overlapping with promoters (±2 kb regions around transcription start sites annotated in GENCODE vM10). These CGI promoters were further tested to see whether they overlapped with the ChIP-seq peaks of H3K4me3 and H3K27me3. Bivalent CGI promoters were defined as CGI promoters that overlapped with both the H3K4me3 and H3K27me3 peaks, whereas active CGI promoters were defined as CGI promoters that overlapped with only the H3K4me3 peaks but not the H3K27me3 peaks.

## Identification of DMVs

To find DMVs, we first identified UMRs (undermethylated regions) and LMRs (low methylated regions) using MethylSeekR (*Burger et al., 2013*) (v1.22.0) with m=0.3 (for P39) or 0.5 (for P0), n=7 and FDR < 0.05. In P39 samples PMDs (partially methylated domains) were excluded from further consideration, and no PMDs were found in P0 samples. DMVs were then defined as UMRs with length ≥5 kb and mean methylation level ≤15%. To compare DMVs identified in the P39 *Dnmt3a* cKO and the control samples, we further grouped these DMVs into six categories, namely consistent (exact same DMV in the two conditions), overhang, cKO unique, control unique, expanded (wider in the cKO), and shrunken (wider in the control) (see illustrations in *Figure 7A*).

For the clustering visualization in *Figure 7C*, we sorted DMVs according to the following criteria: (1) whether they overlap an H3K27me3 DM region; whether they overlap an up- or down-regulated DE gene (with absolute $\log_2$(fold-change) > 0.2); mean mCG > 0.3 in P39 control, P39 cKO; P0 control, or P0 cKO; and finally by the average level of H3K27me3, H3K3me3, and H3K27ac. The 'Fraction DM H3K27me3' shows what fraction of the length of the DMV overlaps a DM H3K27me3 region (called using DiffBind). And we also plotted the mean mRNA logFC for all genes contained within each DMV.

## DMR enrichment around CGI promoter

We used regioneR (*Gel et al., 2016*) (v1.14.0) to test whether two sets of genomic regions had significantly higher numbers of overlaps compared to expected by chance. We used permTest() to perform the permutation test, and used the randomizeRegions() function to generate the shuffled control for 5000 times, where the query regions were randomly placed along the genome independently while maintaining their size. The strength of the association of the two sets of regions was estimated using z-score, the distance (measured in standard deviation) between the expected overlaps in the shuffled control and the observed overlaps, and the p-value was reported. To check if the association was specifically linked to the exact position of the query regions, we used the localZscore() function with window = 5000 and step = 50, which shifted the query regions and estimated how the value of the z-score changed when moving the regions.

## Other tools used in the data analysis

Browser representations were created using AnnoJ (*Lister et al., 2009*). All analyses were conducted in R (v3.5.0), MATLAB 2017a and Python 3. Genomic ranges manipulation was done either with bedtools (*Quinlan and Hall, 2010*) or GenomicRanges (*Lawrence et al., 2013*). To generate randomly shuffled regions used in *Figure 4—figure supplement 3A* and *Figure 5—figure supplement 1A*, we used the sub-command 'shuffle' in the bedtools suite, with the '-excl' parameter to exclude the blacklist regions, the '-noOverlapping' parameter to prevent overlapping shuffled regions, and the optional '-incl' parameter to limit the shuffle regions to reside within the P39 H3K27me3 peak regions (the union of control and cKO peaks). Multiple comparison correction for p-values was performed with the Benjamini-Hochberg FDR method (*Benjamini and Hochberg, 1995*). Results with FDR <

0.05 were considered significant except where stated otherwise. The smoothed lines in *Figure 5D* and *Figure 6—figure supplement 1C* were fitted with a generalized additive model using the 'gam' function in the 'mgcv' R package, with formula = y ~ s(x, bs = 'cs').

## Data access

All sequencing data are available in the Gene Expression Omnibus under accession GSE141587. A genome browser displaying the sequencing data is available at https://brainome.ucsd.edu/annoj/ mm_dnmt3a_ko/.

## Acknowledgements

This work was supported by R01MH112763 to MMB and JRE, and a Kavli Foundation award to MMB, APD, and SBP. JRE is an Investigator of the Howard Hughes Medical Institute. We acknowledge stimulating discussions with Drs. Huda Zoghbi and Laura Lavery. We thank the members of the Salk Biophotonics Core, Dr Uri Manor, Sammy Weiser Novak, and Dr Tong Zhang for their insightful suggestions. We also thank Joseph Chambers and Caitlin Chambers for technical assistance in animal handling, Colleen Heller for technical assistance in behavioral experiments, and Faith Zhang for her initial involvement in the morphometric analysis of dendritic spines. The Waitt Advanced Biophotonics Core Facility at the Salk Institute receives funding from NIH-NCI CCSG: P30 014195 and the Waitt Foundation. The Flow Cytometry Core Facility of the Salk Institute receives funding from NIH-NCI CCSG: P30 014195. The authors have no conflict of interest in relation to the work described here.

## Additional information

### Funding

| Funder | Grant reference number | Author |
|---|---|---|
| National Institute of Mental Health | R01MH112763 | Joseph R Ecker<br>Eran A Mukamel<br>M Margarita Behrens |
| Kavli Foundation | | Antonio Pinto-Duarte<br>Susan B Powell<br>M Margarita Behrens |
| Howard Hughes Medical Institute | | Joseph R Ecker |

The funders had no role in study design, data collection and interpretation, or the decision to submit the work for publication.

### Author contributions

Junhao Li, Conceptualization, Data curation, Formal analysis, Investigation, Methodology, Software, Visualization, Writing – original draft, Writing – review and editing; Antonio Pinto-Duarte, Conceptualization, Data curation, Formal analysis, Funding acquisition, Investigation, Methodology, Validation, Visualization, Writing – original draft, Writing – review and editing; Mark Zander, Data curation, Investigation, Methodology, Validation, Writing – review and editing; Michael S Cuoco, Formal analysis; Chi-Yu Lai, Julia Osteen, Data curation, Investigation, Methodology, Validation; Linjing Fang, Data curation, Investigation, Methodology, Resources, Validation; Chongyuan Luo, Data curation, Formal analysis, Investigation, Methodology, Writing – review and editing; Jacinta D Lucero, Yan Pang, Investigation, Methodology, Resources; Rosa Gomez-Castanon, Joseph R Nery, Investigation, Methodology; Isai Silva-Garcia, Investigation, Methodology, Validation; Terrence J Sejnowski, Resources, Supervision; Susan B Powell, Funding acquisition, Investigation, Methodology, Resources, Supervision, Writing – review and editing; Joseph R Ecker, Conceptualization, Funding acquisition, Project administration, Resources, Supervision, Writing – review and editing; Eran A Mukamel, Conceptualization, Formal analysis, Funding acquisition, Project administration, Resources, Supervision, Visualization, Writing – original draft, Writing – review and editing; M Margarita Behrens, Conceptualization,

Funding acquisition, Investigation, Methodology, Project administration, Resources, Supervision, Writing – original draft, Writing – review and editing

### Author ORCIDs
Junhao Li ⓘ http://orcid.org/0000-0001-6784-3780
Antonio Pinto-Duarte ⓘ http://orcid.org/0000-0002-2215-7653
Mark Zander ⓘ http://orcid.org/0000-0001-8643-1407
Julia Osteen ⓘ http://orcid.org/0000-0001-7058-3297
Linjing Fang ⓘ http://orcid.org/0000-0003-2232-2601
Chongyuan Luo ⓘ http://orcid.org/0000-0002-8541-0695
Terrence J Sejnowski ⓘ http://orcid.org/0000-0002-0622-7391
Joseph R Ecker ⓘ http://orcid.org/0000-0001-5799-5895
Eran A Mukamel ⓘ http://orcid.org/0000-0003-3203-9535
M Margarita Behrens ⓘ http://orcid.org/0000-0002-7168-8186

### Ethics
All animal procedures were conducted in accordance with the guidelines of the American Association for the Accreditation of Laboratory Animal Care and were approved by the Salk Institute for Biological Studies Institutional Animal Care and Use Committee (Protocol number 18-00006).

### Decision letter and Author response
Decision letter https://doi.org/10.7554/eLife.66909.sa1
Author response https://doi.org/10.7554/eLife.66909.sa2

## Additional files

### Supplementary files
• Supplementary file 1. Sequencing metrics for RNA-seq, MethylC-seq, chromatin immunoprecipitation sequencing (ChIP-seq), and genomic DNA for the *Dnmt3a* conditional knockout (cKO) and control samples.

• Supplementary file 2. Gene expression and list of differentially expressed genes in P39 *Dnmt3a* conditional knockout (cKO) and P39 control.

• Supplementary file 3. List of differentially methylated regions (DMRs).

• Supplementary file 4. Known transcription factor motif enrichment in P39 *Dnmt3a* conditional knockout (cKO) differentially methylated regions (DMRs).

• Supplementary file 5. List of transcription factors and chromatin regulators that bind at cis-regulatory regions of the differentially expressed genes in P39 *Dnmt3a* conditional knockout (cKO).

• Supplementary file 6. List of chromatin immunoprecipitation sequencing (ChIP-seq) peaks.

• Supplementary file 7. List of up-regulated H3K27me3 signal regions in P39 *Dnmt3a* conditional knockout (cKO).

• Supplementary file 8. List of enriched Gene Ontology terms for biological process in genes associated with up-regulated H3K27me3 regions in P39 *Dnmt3a* conditional knockout (cKO).

• Supplementary file 9. List of developmental regulated H3K27me3 signal regions in control pyramidal neurons.

• Supplementary file 10. List of DNA methylation valleys (DMVs).

• MDAR checklist

### Data availability
All sequencing data are available in the Gene Expression Omnibus under accession GSE141587. A genome browser displaying the sequencing data is available at https://brainome.ucsd.edu/annoj/mm_dnmt3a_ko/.

The following dataset was generated:

| Author(s) | Year | Dataset title | Dataset URL | Database and Identifier |
|---|---|---|---|---|
| Li J, Pinto-Duarte A, Zander M, Lai C, Osteen J, Fang L, Luo C, Lucero J, Gomez-Castanon R, Nery J, Silva-Garcia I, Pang Y, Powell SB, Ecker JR, Luo C, Mukamel EA, Behrens MM | 2022 | Transcriptomic and epigenetic disruptions in excitatory neurons in Dnmt3a conditional knockout mouse | https://www.ncbi.nlm.nih.gov/geo/query/acc.cgi?acc=GSE141587 | NCBI Gene Expression Omnibus, GSE141587 |

The following previously published dataset was used:

| Author(s) | Year | Dataset title | Dataset URL | Database and Identifier |
|---|---|---|---|---|
| Lister R, Mukamel EA, Nery JR, Urich M, Puddifoot CA, Johnson ND, Lucero J, Huang Y, Dwork AJ, Schultz MD, Yu M, Tonti-Filippini J, Heyn H, Hu S, Rao A, Esteller M, He C, Haghighi FG, Sejnowski TJ, Behrens MM, Ecker JR, Wu JC | 2013 | Global epigenomic reconfiguration during mammalian brain development | https://www.ncbi.nlm.nih.gov/geo/query/acc.cgi?acc=GSE47966 | NCBI Gene Expression Omnibus, GSE47966 |

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

# Appendix 1

## Appendix 1—key resources table

| Reagent type (species) or resource | Designation | Source or reference | Identifiers | Additional information |
|---|---|---|---|---|
| Gene (*Mus musculus*) | *Dnmt3a* | GENCODE | GENCODE:ENSMUSG00000020661 | GENCODE vM10 |
| Genetic reagent (*Mus musculus*) | *Dnmt3a*-floxed | PMID:10555141 | | Backcrossed to C57BL/6 for at least seven generations |
| Genetic reagent (*Mus musculus*) | Nex-Cre; Neurod6-Cre | PMID:17146780 | | Jackson Laboratories Backcrossed to C57BL/6J for 10 generations |
| Genetic reagent (*Mus musculus*) | INTACT | PMID:26087164 | | (B6.129-Gt(ROSA)26Sor tm5(CAG-Sun1/sfGFP) Nat/MmbeJ) |
| Genetic reagent (*Mus musculus*) | *Dnmt3a*-KO; *Dnmt3a* cKO | This paper | | See 'Materials and Methods', section 'Generation of the *Dnmt3a* cKO mice line' |
| Antibody | Anti-NeuN (Mouse monoclonal, Clone A60) | Millipore | MAB377 | (1:1000) |
| Antibody | Anti-DNMT3A (Rabbit polyclonal) | Abcam | Ab2850 | (1:250) |
| Antibody | Anti-H3K27me3 (Rabbit polyclonal) | Active Motif | Cat:#39156; RRID:AB_2636821 | (5 µl) |
| Antibody | Anti-H3K27ac (Rabbit polyclonal) | Active Motif | Cat:#39133; RRID:AB_2561016 | (5 µl) |
| Antibody | Anti-H3K4me3 (Rabbit monoclonal) | Millipore Sigma | Cat:#04–745 | (5 µl) |
| Antibody | Anti-IgG (Mouse unknown clonality) | Jackson ImmunoResearch Labs | Cat#015-000-003; RRID:AB_2337188 | (2 µl) |
| Sequence-based reagent | Cytosine-methylated adapters | Illumina | AD001, AD005 | |
| Sequence-based reagent | TruSeq PCR Primer Mix | Illumina | 20015960, 20015961 | |
| Peptide, recombinant protein | T4 DNA ligase | New England Biolabs | M0202L | |
| Commercial assay or kit | Single-Cell RNA Purification Kit | Norgen | Cat:#51800 | |
| Commercial assay or kit | Ovation RNA-Seq System V2 | NuGEN | Cat:#7102–32 | |
| Commercial assay or kit | KAPA LTP Library Preparation Kit | Roche | Cat:#KK8232 | |
| Commercial assay or kit | Qiagen DNeasy Blood and Tissue kit | Qiagen | Cat:#69504 | |
| Commercial assay or kit | EZ methylation Direct kit | Zymo | Cat:#D5021 | |
| Commercial assay or kit | Accel-NGS 2S Plus DNA Library Kit | Swift Biosciences | Cat:#21024 | |
| Chemical compound, drug | Protease inhibitor | Sigma-Aldrich | Cat:#11836153001 | |
| Chemical compound, drug | Phosphatase inhibitor | Pierce | Cat:#A32957 | |
| Chemical compound, drug | Kapa HiFi Hotstart Readymix | Kapa Biosystems | 07958935001-KK2602 | |
| Software, algorithm | Minianalysis | Minianalysis (https://www.synaptosoft.com/MiniAnalysis/) | RRID:SCR_002184 | |

*Appendix 1 Continued on next page*

*Appendix 1 Continued*

| Reagent type (species) or resource | Designation | Source or reference | Identifiers | Additional information |
|---|---|---|---|---|
| Software, algorithm | pCLAMP | Molecular Devices (https://www.moleculardevices.com/products/software/pclamp.html) | RRID:SCR_011323 | |
| Software, algorithm | Reconstruct | Synapse Web (https://synapses.clm.utexas.edu/tools/reconstruct/reconstruct.stm) | RRID:SCR_002716 | |
| Software, algorithm | Activity Monitor | MED Associates (https://www.med-associates.com/product/activity-monitor/) | RRID:SCR_014296 | |
| Software, algorithm | ANY-maze | San Diego Instruments (https://sandiegoinstruments.com/product/any-maze/) | RRID:SCR_014289 | |
| Software, algorithm | FastQC | Babraham Bioinformatics (https://www.bioinformatics.babraham.ac.uk/projects/fastqc/) | RRID:SCR_014583 | v0.11.8 |
| Software, algorithm | Trim Galore | Babraham Bioinformatics (https://www.bioinformatics.babraham.ac.uk/projects/trim_galore/) | RRID:SCR_011847 | v0.5.0 |
| Software, algorithm | Cutadapt | DOI:10.14806/ej.17.1.200 | RRID:SCR_011841 | v1.16 |
| Software, algorithm | STAR | PMID:23104886 | RRID:SCR_004463 | v2.5.1b |
| Software, algorithm | RSEM | PMID:21816040 | RRID:SCR_013027 | v1.2.30 |
| Software, algorithm | edgeR | PMID:19910308 | RRID:SCR_012802 | v3.24.1 |
| Software, algorithm | SQuIRE | PMID:30624635 | | |
| Software, algorithm | clusterProfiler | PMID:22455463 | RRID:SCR_016884 | v3.10.0 |
| Software, algorithm | GATK | PMID:21478889 | RRID:SCR_001876 | v3.7 |
| Software, algorithm | BWA | PMID:19451168 | RRID:SCR_010910 | v0.7.13-r1126 |
| Software, algorithm | Picard | Broad (https://broadinstitute.github.io/picard/) | RRID:SCR_006525 | V2.10.6 |
| Software, algorithm | BEDTools | PMID:20110278 | RRID:SCR_006646 | v2.27.1 |
| Software, algorithm | methylpy | methylpy (https://github.com/yupenghe/methylpy) | | v1.3.2 |
| Software, algorithm | bowtie2 | PMID:22388286 | RRID:SCR_016368 | v2.3.0 |
| Software, algorithm | MatchIt | DOI:10.18637/jss.v042.i08 | | v3.0.2 |
| Software, algorithm | BART | PMID:29608647 | | |
| Software, algorithm | HOMER | PMID:20513432 | RRID:SCR_010881 | v4.8.3 |
| Software, algorithm | ENCODE Transcription Factor and Histone ChIP-Seq processing pipeline | PMID:22955991 | RRID:SCR_021323 | v1.1.6 |
| Software, algorithm | samtools | PMID:19505943 | RRID:SCR_002105 | v1.2 |
| Software, algorithm | SICER | PMID:19505939 | RRID:SCR_010843 | |
| Software, algorithm | DiffBind | PMID:22217937 | RRID:SCR_012918 | v2.10.0 |

*Appendix 1 Continued on next page*

*Appendix 1 Continued*

| Reagent type (species) or resource | Designation | Source or reference | Identifiers | Additional information |
|---|---|---|---|---|
| Software, algorithm | DESeq2 | PMID:25516281 | RRID:SCR_015687 | v1.22.1 |
| Software, algorithm | GREAT | PMID:20436461 | RRID:SCR_005807 | v3.0.0 |
| Software, algorithm | MethylSeekR | PMID:23828043 | RRID:SCR_006513 | v1.22.0 |
| Software, algorithm | regioneR | PMID:26424858 | | V1.14.0 |
| Software, algorithm | R | R Project for Statistical Computing (https://www.r-project.org/) | RRID:SCR_001905 | v3.5.0 |
| Software, algorithm | MATLAB | MathWorks (https://www.mathworks.com/products/matlab.html) | RRID:SCR_001622 | v2017a |
| Software, algorithm | Python | Python Programming Language (https://www.python.org/) | RRID:SCR_008394 | v3.x |

