## [Editor Report]

In this manuscript the authors conditionally knock out the DNA methyltransferase Dnmt3a in developing excitatory cortical neurons to determine the consequences for chromatin regulation, gene expression, and neuron function. As expected they find widespread loss of DNA methylation but also an increase in histone methylation (H3K27me3) at many similar regions of the genome, which they propose may be a mechanism of functional compensation. Overall this study offers new insights into the gene regulatory and neuronal cellular functions of an important chromatin regulatory process.

---

## [Decision Letter]

**Decision letter after peer review:**

Thank you for submitting your article "Dnmt3a knockout impairs synapse maturation and is partly compensated by repressive histone modification H3K27me3" for consideration by *eLife*. Your article has been reviewed by 3 peer reviewers, one of whom is a member of our Board of Reviewing Editors, and the evaluation has been overseen by a Reviewing Editor and Lu Chen as the Senior Editor. The following individual involved in review of your submission has agreed to reveal their identity: Alvaro Rada-Iglesias (Reviewer #3).

Essential revisions:

1) The reviewers all felt that claims regarding the functional compensation by H3K27me3 for the loss of Dnmt3a were too strongly stated. The reviewers discussed this point extensively and decided that both the title and the sections of the paper that describe and discuss these data should be changed to reflect that the replacement of the marks has been observed but that the functional relevance of this change has not been directly tested. The reviewers appreciate that functional testing of this hypothesis is well beyond the scope of the current manuscript – only text revisions are required.

2) The reviewers strongly suggest (but do not absolutely require) that the authors perform bioinformatic analysis of the H3K27me3 compensation against gene expression in the Dnmt3a cKO in order to provide experimental evidence for the claim of functional compensation. One of the reviewers also offered a very thorough list of other possible analyses (especially regarding studies of enhancer marks) that would strengthen the study, however these are entirely at the discretion of the authors to consider.

3) The reviewers have described below specific examples where addition of further technical details would strengthen the manuscript and we ask the authors to expand on these experimental points in the text to help the readers evaluate the study. In particular all of the reviewers expressed some concern about the use of only two replicates, and we had a long discussion of whether a minimum of three should be required. It was not entirely clear in every case whether each replicate was obtained from only a single mouse or whether the samples might be pooled (which would strengthen the argument for using only 2). In either case, in the end we felt the manuscript was strong enough that we did not feel it was necessary for the authors to add more replicates, but we would like the authors to address in the text any limitations of the study that readers may need to keep in mind with the use of duplicates as opposed to larger replicate numbers.

*Reviewer #1 (Recommendations for the authors):*

1) Because of the way the authors present the examples and the comparisons, it is unclear if the compensatory upregulation of H3K27me3 occurs de novo, or whether regions that are previously methylated with H3K27me3 are the regions where this compensation occurs. The latter could suggest this might be a consequence of having more substrate for histone methylation – and may not actually serve any function. This is particularly important to delineate as there are several other DMV clusters that are resistant to changes in gene expression that do not have this compensatory upregulation of H3K27me3 – like clusters 4, 5 and 6 in Figure 8.

2) It would be more narratively and biologically significant if the authors could connect their data analysis and observations about compensatory H3K27me3 upregulation and changes to gene expression in the context of the developing cortical neurons, or the behavioral phenotypes observed in the introductory figures.

3) The inferences made in Figure 8 are a little inconsistent with those described earlier. For example, in Figure 5A the authors describe the overrepresentation of predicted PRC2-TF binding motifs at upregulated and downregulated genes during Dnmt3a-cko, while later discussions move toward describing PRC2-mediated regulation as the reason for resistance to changes in gene expression due to the cko. Additionally, H3K27me3 peaks in Figure 5B do not appear to be all that different between the control and cko mouse at Mab21l2 – diluting the punchline of the figure.

4) Figure 8 is difficult to read.

*Reviewer #2 (Recommendations for the authors):*

As described in the public comments, this study provides interesting new insights into organismal and cellular consequences of DNMT3A conditional knockout in excitatory neurons and presents novel effects on H3K27me3 in these mice. I have several concerns I think would be important to address before publication in *eLife* however. Here I elaborate in specific detail:

1. The analysis and core conclusions about changes in histone modifications in the study are based on n=2 experiments. While it is potentially beyond the scope of a revision to repeat additional replicates for all analyses that are carried out, it would substantially strengthen the study to perform additional replicates of H3K27me3 ChIP in order to rigorously show that the core finding regarding H3K27me3 changes is robust to sample-to-sample technical variability. This is particularly important given the subtle magnitude of changes that are described.

2. Though the authors provide strong characterization of developmental changes of H3K27me3 and show changes in this mark in response to loss of DNMT3A, their analyses fall short of definitively showing that this is a true functionally compensatory effect. Ideally, an experimental analysis would be done to test this by blocking H3K27me3 build up in DNMT3A mutants and determining if de-compensated gene expression changes occur. Given that this is outside the scope of a revision however, additional computational approaches could be employed to better test the hypothesis:

a. Figures 6E and S10A present the analysis that is most relevant to whether accumulation of K27me3 in the cKO impacts DEG and therefore may play a compensatory role. However, additional analysis could more decisively test the hypothesis. For example, do non-dysregulated genes containing DMRs show less K27me3 accumulation than down-regulated genes, but more K27me3 accumulation than upregulated genes? The authors could include a boxplot of a well-matched set of non-DE genes with overlapping DMRs in panel 6E to test this. A more powerful analysis, however, might be to examine a large set of genes from the genome that are selected to contain similar numbers of DMRs (in order to control for relative potential effects of demethylation) and ask if changes in K27me3 correspond with up-regulation (low K27me3 accumulation), no dysregulation (intermediate accumulation) or downregulation (high K27me3 accumulation).

b. On p19 the authors state that "P39 cKO DMRs were significantly enriched in peaks and DM regions of H3K27me3 (28.1% overlapped H3K27me3 peaks in cKO, p<1e-100) (Figure 6B)." However, 6B only presents the overlap with peaks in each genotype as an odds ratio and doesn't make it clear how the enrichment of DMRs in DM regions was assessed. This appears to be assessed to some degree in S8F (note the S8 figure legend appears to be wrong for this panel), but no analysis is presented to control for the general tendency of DMRs and H3K27me3 peaks to overlap. It seems imperative, given that DMRs already overlap with H3K27me3 peaks, that a resampling approach is used comparing DM peaks to non-DM K27me3 peaks that are matched to have similar wild-type H3K27me3 signal and size as DM peaks. Likewise, what would the changes in H3K27me3 signal presented in 6C look like at non-DMR regions that contain similar baseline wild-type H3K27me3 signal? Without these analyses it is not clear that DMRs are particularly unique in showing increased K27me3 signal.

c. The increase in H3K27me3 in the cKO at the mab21l2 locus at P0 (figure 5B), before methylation has or has not been established, suggests that H3K27me3 may change before methylation differences, rather than in reaction to changes in methylation. Is this a common effect seen across other genes? How does this impact the idea that compensation occurs due to lack of methylation? Related to this issue, on p.19 no specific plot is referenced to support the statement that "There was no corresponding increase of H3K27me3 at these DMRs in newborn (P0) or fetal (E14) neurons." A quantitative analysis as shown in 6C but for the E14 and P0 time-points could assess this directly.

d. Why were developmental DMs called between E14 and P39 while DMRs were called between P0 and P39? Developmental comparisons would be more justified if both used P0 and P39.

e. In the absence of additional data supporting H3K27me3 accumulation as a functional compensation mechanism for loss of DNA methylation, the title of Figure 6 should not so strongly claim that H3K27me3 changes compensate for mC loss.

3. While initially assessed by the authors, mCH changes are not integrated into the analysis of H3K27me3 changes. Previous studies by this group and others have suggested the importance of mCH in gene regulation within specific neuron populations and, given the interest to the field, more thorough analysis is warranted regarding this neuronal methylation and H3K27me3.

a. Given the association, even if limited, between differential mCH loss and gene dysregulation in the cKO, it is relevant to examine how changes in K27me3 relate to, or compensate for, loss of mCH. For example, do genes that are not dysregulated in the cKO, and normally have high mCH (which is lost in the cKO) show higher increases in K27me3 compared to non-dysregulated genes that have low mCH (and therefore lose less mCH)? Reciprocally, one could ask if a population of genes selected to have similar, high mCH loss in the cKO, show differential gene dysregulation as a function of differential K27me3 accumulation.

b. What are the changes in mCH at DM H3K27me3 sites? Since mCG and mCH loss are correlated, one might expect that these are sites that normally also contain high mCH. It might not be feasible to call mCH DMRs given the technical limitations, but quantification of this signal can be done at DMs and DMRs, and the potential for it to contribute to these effects could be discussed.

4. The authors present evidence of PRC2 binding at DEGs (Figure 5A) as a motivation to assess H3K27me3 in their mice. However, they then observe changes in H3K27me3 that they contend do not occur under normal conditions and propose that these changes are an underlying mechanism that blocks gene dysregulation in these mice. How are we to reconcile this original rationale of a normal function for PRC2 at dysregulated genes with the model of de novo compensatory H3K27me3 accumulation? Some explanation by the authors of how these separate notions may relate to one another would be helpful.

5. Some effects on H3K27ac are observed at DMRs, but the authors ignore the prospect that these effects could drive changes in gene expression. Previous groups have shown significant changes in H3K27ac in DNMT3A models, and some of the data the authors present suggest that this could also explain some of the observations in their model.

a. While DM K27ac peaks are not identified in the cKO, in figure 6B there is a similar enrichment of DMRs to associate with K27ac peaks in the cKO vs control as there is for K27me3. In addition, there is a similar increase in odds ratio of enrichment of K27ac peaks to the of K27me3 at DMRs for the cKO in figure 6B. The analysis in S8F-G that is referenced when referring to K27ac is either missing (S8G) or is not showing data for K27ac (S8F). The authors should at least comment in the Results section that these signals exist and may be significant for gene regulation.

b. Are two biological replicates sensitive enough to call differences in H3K27ac? In the context of a small number of replicates and the subtle magnitude of changes in gene expression that are observed, negative results for differential peaks should be interpreted with caution. In contrast, the larger peaks observed in K27me3 may have led to increased ability to call differential peaks.

6. Though the characterization of DNMT3A cKO in excitatory neurons is novel, if the authors are claiming that these neurons specifically are driving behavioral phenotypes then further contextualization and characterization of the extent of conditional deletion is needed.

a. It would be useful to justify the rationale for investigating excitatory neurons, and how using this class of neurons can uncover novel findings. Existing studies of DNMT3A cKO have shown behavioral and epigenomic effects in pan-neuronal and inhibitory neuron conditional knockouts. Comparisons between these models and discussion of differences could offer strong insights into what unique circuits may be most disrupted by DNMT3A cKO in excitatory neurons.

b. If the authors are claiming that behaviors and phenotypes are the result of specifically knocking out DNMT3A in all excitatory neurons, further evidence should be provided that the Cre-line is being expressed sufficiently and specifically in excitatory neurons, and that their genomic data is faithfully capturing the majority of excitatory cells while excluding non-excitatory cells.

7. The manuscript would be further improved by clarification of some methods and statistics, and modifications in data presentation:

a. The authors should provide information on the statistics performed on the behavior data. Are there significant differences between sexes, or significant sex by genotype interactions for these behavioral tasks? A two-way ANOVA using genotype and sex would be useful in understanding the impact of these variables. There appears to be many effects that are sex-specific, yet this appears to be overlooked in the text. Furthermore, if there are robust sex-specific effects then the subsequent molecular and genomic analyses should be done using both sexes, and comparisons could be done to explain the molecular underpinnings of these differences.

b. The authors might also consider showing preference index when analyzing social approach data when comparing genotypes.

c. What is the background set of genes for 2B-C? Synaptic genes are enriched in both up- and down-regulated gene lists. How is gene expression normalized? Is there a discovery bias for more highly expressed genes?

d. What value was used to identify equally expressed unchanged genes in figure 4B, TPM? Identifying equally expressed genes can be complicated for nuclear RNA where unprocessed RNAs contribute differentially to total counts based on the length of the gene. In addition, previous analysis of mCH enrichment at dysregulated genes in inhibitory neuron-specific KO of DNMT3A also found DEGs to be differentially methylated based on direction of change but showed that all other genes were more lowly methylated than these two groups (Lavery et al. 2020). Here the authors see the same relationship between DEG groups but find other genes to be higher in mCH rather than lower. Do the authors have any perspective on why these differences are observed?

e. On p22 the authors state "We found that developmental peaks (DevLoss and DevGain) were relatively unaffected by the cKO(H3K27me3 = 0.11, -0.20 respectively, in units of log10(CPM+1)), whereas Group cKO had ten-fold larger mean effect (H3K27me3 = 1.12 respectively)." Given the y scales in figures 7E and S10H it is unclear how they derived a ∆ value of 1.12 on a log 10 scale for Group cKO (e.g. in S10H the median values look much less 1 unit different on a log10 scale). Please clarify the actual magnitude of effect sizes.

f. The authors describe the association between gene body mCH and gene expression changes as only accounting for 1% of the variance. It would be useful to have a point of reference to understand if this is a low value or a high value in comparison to other potential predictors of gene expression changes. For example, how much does the proposed overcompensation by H3K27me3 accumulation contribute to the genome-wide variance in gene expression? In addition, is it the case that variance in the measured fold-changes will be mitigated by addition of more replicates to more accurately estimate true fold-changes? If this is the case, it should be noted.

g. Using different color schemes would be helpful in 8A and 8B to avoid confusion.

h. How was clustering analysis done in 8D? What do the values for K27me3 Diff. Mod and DEG enrichment indicate (i.e. what does the scale from 0 to 0.02 represent)?

*Reviewer #3 (Recommendations for the authors):*

– The authors delete exon 19 of Dnmt3a. I think they should provide a more thorough description of what this deletion is expected to cause: loss of function due to premature stop codon, loss of enzymatic activity, etc. Moreover, it would be desirable to show by immunostaining whether DNMT3A protein is specifically lost in excitatory neurons.

– Regarding the DMRs identified in DNMT3A KO neurons, the authors should use the generated profiles for H3K27ac in P0 and P39 WT neurons to determine which DMRs overlap enhancers that are active in any of these two neuronal stages. This is more relevant than overlaps with enhancers that could be active in any given cell type. Most importantly, the authors should evaluate the impact the loss of DNA methylation can have on enhancer activity by measuring H3K27ac levels and eRNAs in DNMT3A KO neurons. Measuring eRNA levels can be particularly relevant, as recent evidences suggest that is a better marker of enhancer activity than H3K27ac.

– The authors should aim at performing experiments that address the proposed compensatory role for PRC2 and H3K27me3. I understand that generating a KO for PRC2 could be rather complicated, but perhaps the authors could consider using new drugs that act as very specific inhibitors against the enzymatic activity of PRC2.

---

## [Author Response]

Essential revisions:1) The reviewers all felt that claims regarding the functional compensation by H3K27me3 for the loss of Dnmt3a were too strongly stated. The reviewers discussed this point extensively and decided that both the title and the sections of the paper that describe and discuss these data should be changed to reflect that the replacement of the marks has been observed but that the functional relevance of this change has not been directly tested. The reviewers appreciate that functional testing of this hypothesis is well beyond the scope of the current manuscript – only text revisions are required.

We agree with the Reviewers that it is important to clearly define the conclusions we can draw from our data, as well as their limitations. Indeed, our data demonstrate that loss of *Dnmt3a*-dependent gain of methylation during the perinatal period leads to specific changes in H3K27me3, but they do not fully clarify the functional significance of these processes or show that Polycomb associated marks fully compensate for reduced DNA methylation.

To clarify this, we have revised the title of the paper: Dnmt3a knockout in excitatory neurons impairs postnatal synapse maturation and increases the repressive histone modification, H3K27me3

We have further removed the following phrase from the Abstract: “partially compensating for the loss of DNA methylation.” We have limited to the Discussion section our comments about a potential compensatory role for the gain of H3K27me3.

2) The reviewers strongly suggest (but do not absolutely require) that the authors perform bioinformatic analysis of the H3K27me3 compensation against gene expression in the Dnmt3a cKO in order to provide experimental evidence for the claim of functional compensation. One of the reviewers also offered a very thorough list of other possible analyses (especially regarding studies of enhancer marks) that would strengthen the study, however these are entirely at the discretion of the authors to consider.

We appreciate the suggestions from the reviewers. We have added new analyses to further support our claim, including comparisons of the changes in H3K27me3 in different groups of genes (shown in Figure 5E and Figure 5 — figure supplement 2A). These are fully described below, in the response to Reviewer #2 (comment 2a).

3) The reviewers have described below specific examples where addition of further technical details would strengthen the manuscript and we ask the authors to expand on these experimental points in the text to help the readers evaluate the study. In particular all of the reviewers expressed some concern about the use of only two replicates, and we had a long discussion of whether a minimum of three should be required. It was not entirely clear in every case whether each replicate was obtained from only a single mouse or whether the samples might be pooled (which would strengthen the argument for using only 2). In either case, in the end we felt the manuscript was strong enough that we did not feel it was necessary for the authors to add more replicates, but we would like the authors to address in the text any limitations of the study that readers may need to keep in mind with the use of duplicates as opposed to larger replicate numbers.

We appreciate this concern, and we agree that it is important to show that findings are replicable across multiple independent samples.

When tracing back the animal IDs to create a table of the number of animals included in each sample, we discovered an error in the dataset. Although we intended to use independent pools of mice for each biological replicate, we found that due to an experimental tracking error some of the replicates contained tissue from overlapping pools of animals. In particular, the P39 *Dnmt3a* cKO group (pooled from 3 mice) was split into two technical replicates but mislabeled as two biological replicates and used as such in the RNA-seq and MethylC-seq analyses. For this reason, for this resubmission we decided to repeat our experiments and to generate new RNA-seq data and MethylC-seq data for both P39 *Dnmt3a* cKO and control mice. Each condition now has two independent, newly-generated biological replicates, each pooled from two mice. We have added the information about the number of animals contributing tissue to each sample (including P0 MethylC-seq data and all ChIP-seq data which are not re-sequenced) in the “num_pooled_animals” column in Supplementary File 1. Most of our sequencing data come from tissue samples pooled from two mice, with the exception of P0 MethylC-seq data. For that time point, we used 6 control and 2 cKO samples that each came from a single mouse.

To show that the new dataset largely agrees with the previously collected data, we plotted the correlation across replicates from the two batches computed using mCG and mCH in 10kb genomic bins for MethylC-seq data, and gene expression for RNA-seq data (see Author response image 1). The molecular profiles are largely consistent across the two batches (Spearman correlation > 0.80; note that mCH in *Dnmt3a* cKO samples are essentially absent, so those samples were excluded from the mCH analysis). Moreover, the consistency across biological replicates in the new batch is also high (r > 0.93). We have also added plots in the manuscript to demonstrate the consistency across the replicates for RNA-seq (Figure 3 — figure supplement 1A), MethylC-seq (Figure 3 — figure supplement 3A), and ChIP-seq (Figure 6 — figure supplement 2).

**Author response image 1. sa2fig1:** Heatmap to show the Spearman correlations across the control and *Dnmt3a* cKO samples from the two batches of data (new vs old). The correlations coefficients were computed using CG methylation levels in 10Kb genomic bins (left), CH methylation levels in 10Kb genomic bins (middle), and gene expression (log_10_TPM, right).

As is often the case for RNA-seq data, we found differences between the new and old batches (i.e. batch effects) that cannot be ignored (see Author response image 2). Due to the mislabeling of technical replicates as biological replicates in our previous analysis, our original submission likely had inflated statistical significance for detecting differentially expressed genes. Using the new batch of RNA-seq data that includes true biological replicates, we detected fewer significant DE genes under the same FDR < 0.05 criteria (70 genes compared to the previous 1720 genes). However, the fold-changes of these DE genes are in general concordant across the two batches (see Author response image 2). This new dataset and analysis replicate and validate the disruption of transcriptome we observed earlier. The smaller number of differentially expressed genes we now report is consistent with the fact that statistical significance was inflated in our previous analysis by the non-independence of the replicate samples.

**Author response image 2. sa2fig2:** (A) PCA plot using log_10_TPM gene expression in the two batches of RNA-seq samples. (B) Scatter to show the consistency of gene expression fold-changes (*Dnmt3a* cKO vs. control) across the two batches of RNA-seq samples using significant DE genes detected in the new batch (left) and significant DE genes detected in the old batch (right).

Confidence in our differential expression findings is also supported by the strong correspondence we now find between genes that lose mCH and those that increase mRNA expression (Figure 3D-E). This gives additional confidence that the differential expression analysis identifies reliable and consistent effects in the *Dnmt3a* cKO.

We apologize for the confusion caused by the replacement of these new sequencing datasets. We hope that we have demonstrated that the new data largely agrees with the old, and that most of our conclusions are still supported by the new data albeit with reduced statistical power. In the revised manuscript, all results generated using P39 RNA-seq and MethylC-seq data were updated with the new datasets. Nevertheless, some analysis is no longer feasible due to fewer detected DE genes, and we have modified the manuscript accordingly. In the following responses, we address each reviewers’ specific concerns.

Reviewer #1 (Recommendations for the authors):1) Because of the way the authors present the examples and the comparisons, it is unclear if the compensatory upregulation of H3K27me3 occurs de novo, or whether regions that are previously methylated with H3K27me3 are the regions where this compensation occurs. The latter could suggest this might be a consequence of having more substrate for histone methylation – and may not actually serve any function. This is particularly important to delineate as there are several other DMV clusters that are resistant to changes in gene expression that do not have this compensatory upregulation of H3K27me3 – like clusters 4, 5 and 6 in Figure 8.

We appreciate the reviewer’s suggestion and apologize for the confusion. To address this question, we have added the following analysis (Figure 4 — figure supplement 3A) to compare the baseline H3K27me3 levels in the differentially modified regions with their shuffle controls:

Here two types of random shuffle controls were used. In the first shuffle control (the middle two violins), we randomly shuffled the H3K27me3 DM regions in the whole genome, keeping the peak sizes and the chromosomes where the peaks reside in same as the original peaks. In the second set (the rightmost two violins) we required the same constraints as the first set but further restricted the shuffles must be within the H3K27me3 peaks (the union of control and cKO peaks). We can see that the regions with up-regulation of H3K27me3 (the leftmost two violins), had an intermediate level of H3K27me3 (median ~8 RPKM) in the control samples to begin with, which is higher than the level in random shuffles in the genome but lower than the level in random shuffles in H3K27me3 peaks. Therefore, these regions with upregulation of H3K27me3 after *Dnmt3a* cKO include regions that were previously methylated with H3K27me3 in the control samples, and we presume such a medium but non-zero level of H3K27me3 is fine-tunable and hence can act as a “failsafe” triggered after the original repression mechanism, DNA methylation, was disrupted. Of course, it is important to note that our ChIP-seq data come from a heterogeneous pool of pyramidal neurons, and it is possible that the quantitative increase in the H3K27me3 level corresponds to the genuine gain of de novo histone methylation in a particular subpopulation of excitatory cells.

We can also reach a similar conclusion in our updated DMV analysis (now in Figure 7C). C1 clusters are DMVs that highly overlap with the H3K27me3 DM regions, and they have an intermediate level of H3K27me3 (~8 RPKM) that is not as high as those in cluster 6 (~20 RPKM) but still higher than those in clusters 5, 7 and 8 (~0 RPKM). As a comparison, regions in cluster 6 were under tight regulation with high levels of H3K27me3 and a low level of DNA methylation (~10% mCG); genes within these DMVs were mostly not expressed. Therefore, losing the small amount of DNA methylation in these regions did not alter gene expression. As for the regions in clusters 5, 7 and 8, they are active with both low DNA methylation and H3K27me3, and hence they are not affected by the cKO.

We have added these results to the Result section of the manuscript:

“These DM regions have a medium but non-zero level of H3K27me3 in the P39 control (higher than random shuffles across the whole genome but lower than random shuffles within the peak regions, Figure 4 — figure supplement 3A), and hence presumably fine-tunable after Dnmt3a cKO.”

2) It would be more narratively and biologically significant if the authors could connect their data analysis and observations about compensatory H3K27me3 upregulation and changes to gene expression in the context of the developing cortical neurons, or the behavioral phenotypes observed in the introductory figures.

We agree that linking the effects of the cKO on adult neurons with developmental patterns of epigenetic regulation would be valuable. Our study includes developmental ChIP-seq profiles of H3K27me3, including three time points (E14, P0 and P39; see Figure 5B-C). These data show that the majority of the changes in H3K27me3 emerge in the adult neurons (P39) and are not present at earlier developmental time points (Figure 4 — figure supplement 1D). Moreover, we related the patterns of regulations at DMVs with their developmental history of H3K27me3 (Figure 7).

We have now added a new analysis to address the relationship between upregulated H3K27me3 signal and gene expression changes at P39 (see Figure 5 — figure supplement 2C and also our replies to reviewer #2, comment 3a).

Unfortunately, we did not profile gene expression after cKO in development so we can’t examine these observations in the context of development. Although we agree that this would be of interest, we believe that this question would be better addressed using single-cell transcriptomics (scRNA-seq) to disentangle the contribution from diverse subtypes of excitatory pyramidal neurons. We, therefore, plan to apply single-cell methods in the future to address this question.

3) The inferences made in Figure 8 are a little inconsistent with those described earlier. For example, in Figure 5A the authors describe the overrepresentation of predicted PRC2-TF binding motifs at upregulated and downregulated genes during Dnmt3a-cko, while later discussions move toward describing PRC2-mediated regulation as the reason for resistance to changes in gene expression due to the cko.

The BART results for up-regulated and down-regulated DE genes (now in Figure 4A) are bioinformatics predictions that motivated us to look into Polycomb repression via H3K27me3 profiling. With this rationale, we then performed the ChIP-seq analysis of H3K27me3 in the control and cKO neurons. We agree with the reviewer that there is not a clear explanation for the enrichment of PRC2 related sequence motifs at both up- and down-regulated genes. In light of this, and in response to other reviewer comments, we have significantly softened our interpretation of the direct link (“compensation”) between loss of DNA methylation and increased H3K27me3. Instead, we now report that increased H3K27me3 occurs preferentially in regions that lose DNA methylation, but we do not draw a causal link or interpret these data in terms of direct functional compensation.

Additionally, H3K27me3 peaks in Figure 5B do not appear to be all that different between the control and cko mouse at Mab21l2 – diluting the punchline of the figure.

Regarding the size of the difference in H3K27me3 at the Mab21l2 locus, we appreciate that it can be hard to judge the magnitude of the differential enrichment of the chromatin modification (due to the normalized scale to make it comparable across E14, P0, and P39) in the browser view (now in Figure 4B). The quantitative analysis of differential binding at this locus showed that it was highly significant (tested with DEseq2, fold-change = 2.41, FDR=1.33e-4). We have added a shaded box in Figure 4B to highlight the H3K27me3 signal differences between the P39 control and cKO mouse at the Mab21l2 locus, and also a new plot in Figure 4C to show the quantification of the increase in H3K27me3 ChIP-seq signal in each replicate at this locus.

We have also added analysis to show the increases of H3K27me3 were also enriched in non-DE genes with the largest magnitude of mCH (see Figure 5 — figure supplement 2B and also our replies to reviewer #2, comment 3a).

4) Figure 8 is difficult to read.

We have re-organized the figure (now Figure 7) to make it easier to read.

Reviewer #2 (Recommendations for the authors):As described in the public comments, this study provides interesting new insights into organismal and cellular consequences of DNMT3A conditional knockout in excitatory neurons and presents novel effects on H3K27me3 in these mice. I have several concerns I think would be important to address before publication in eLife however. Here I elaborate in specific detail:1. The analysis and core conclusions about changes in histone modifications in the study are based on n=2 experiments. While it is potentially beyond the scope of a revision to repeat additional replicates for all analyses that are carried out, it would substantially strengthen the study to perform additional replicates of H3K27me3 ChIP in order to rigorously show that the core finding regarding H3K27me3 changes is robust to sample-to-sample technical variability. This is particularly important given the subtle magnitude of changes that are described.

We understand the reviewer’s concern regarding the limited number of biological replicates. To reduce variability due to individual differences, each of our ChIP-seq samples included pooled tissue from 2 mice. This information is now included in the

“num_pooled_animals” column in Supplementary File 1 and explained in the main text:

“To experimentally address this, we performed chromatin immunoprecipitation sequencing (ChIP-seq) in excitatory neurons at embryonic day 14 (E14) and postnatal days 0 and 39 to measure trimethylation of histone H3 lysine 27 (H3K27me3), a repressive mark whose deposition is catalyzed by PRC2 and is important for transcriptional silencing of developmental genes. In P39 neurons, we also measured two histone modifications associated with active chromatin: H3K4me3 (trimethylation of histone H3 lysine 4, associated with promoters) and H3K27ac (acetylation of histone H3 lysine 27, associated with active promoters and enhancers) (Heinz et al., 2015). For each mark, we performed sequencing on two independent samples, each of which used pooled tissue from two mice.”

We have added additional analyses showing the consistency of the ChIP-seq signal in genomic bins and peak regions across biological replicates. These results are now shown in Figure 6 — figure supplement 2 and we have added descriptions of these results in the manuscript as follows:

“We also examined differentially modified (DM) regions at P0 vs. E14 and P0 vs. P39 (Figure 6 — figure supplement 1A). However, due to greater biological variability at the perinatal time point, the two replicate ChIP-seq datasets at P0 were less consistent than the E14 and P39 samples (Figure 6 — figure supplement 2). As a result, our data were not well powered to detect changes at P0 and we chose to focus on the E14 vs. P39 DM regions as sites of developmental chromatin remodeling.”

2. Though the authors provide strong characterization of developmental changes of H3K27me3 and show changes in this mark in response to loss of DNMT3A, their analyses fall short of definitively showing that this is a true functionally compensatory effect. Ideally, an experimental analysis would be done to test this by blocking H3K27me3 build up in DNMT3A mutants and determining if de-compensated gene expression changes occur. Given that this is outside the scope of a revision however, additional computational approaches could be employed to better test the hypothesis:

We agree that additional experiments that block H3K27me3 in *Dnmt3a* cKO animals are required to establish a causal role for the compensating effect. However, as noted by the Reviewer, such experiments are beyond the scope of this study. Instead, we have performed additional bioinformatic analyses to better test the hypothesis according to the reviewer’s suggestion.

As suggested by the reviewer, we used box plots to compare the gain of H3K27me3 in different groups of genes. These analyses (now in Figure 5E and Figure 5 — figure supplement 2A) compare up- and down-regulated genes with the control gene set chosen to have matching expression levels in the control samples. Moreover, we separately analyzed genes that overlap DMRs and those which do not overlap DMRs; the corresponding control genes were likewise grouped by whether or not they overlap DMRs. (See Methods and Figure 3 — figure supplement 4B on how we selected the expression-matched non-DE genes).

This analysis shows that down-regulated DE genes with overlapping DMRs (the first box in Figure 5E, upper panel) and non-DE genes selected with matched expressions of up-regulated DE genes (the fourth box in Figure 5E, upper panel) show a small but significant median increase of H3K27me3 in the cKO (Wilcoxon rank-sum test; asterisks in the middle of each boxplot denotes p-value: *, p < 0.05). When we considered a larger set of DE genes with a more relaxed threshold (FDR<0.2, Figure 5 — figure supplement 2A), we observed that down-regulated DE genes containing DMRs accumulate significantly more H3K27me3 than non-DE genes containing DMRs. By contrast, up-regulated genes had no significant accumulation of H3K27me3 and were not significantly different from the control genes (Figure 5 — figure supplement 2A). No such differences were observed between DE and non-DE genes without overlapping DMRs (Figure 5E lower panel and Figure 5 — figure supplement 2A). These results suggest that the effect of H3K27me3 is likely specific to genes containing DMRs and the effect is stronger in the down-regulated DE genes.

We have added descriptions of these results in the manuscript as follows:

“We further examined whether the increased H3K27me3 could account for the reduced expression of some genes in the Dnmt3a cKO neurons. We found that down-regulated genes that overlap DMRs and the non-DE genes selected with matched expressions of up-regulated DE genes that overlap DMRs showed a small but significant median increase of H3K27me3 in the cKO (Wilcoxon rank-sum test p-value < 0.05; Figure 5E upper panel). When we considered a larger set of DE genes with a more relaxed threshold (FDR<0.2), we observed that down-regulated DE genes containing DMRs accumulate significantly more H3K27me3 than non-DE genes containing DMRs and up-regulated genes containing DMRs (Wilcoxon rank sum test p = 0.0029, Figure 5 — figure supplement 2A). By contrast, up-regulated genes had no significant accumulation of H3K27me3 and were not significantly different from the control genes (Figure 5 — figure supplement 2A). No such differences were observed between DE and non-DE genes without overlapping DMRs (Figure 5E lower panel and Figure 5 — figure supplement 2A). These results suggest that the effect of H3K27me3 is likely specific to genes containing DMRs and the effect is stronger in the downregulated DE genes, which may partially explain the fact that 24 genes were significantly down-regulated after the loss of repressive DNA methylation in the Dnmt3a cKO (Figure 2A-B).”

a. Figures 6E and S10A present the analysis that is most relevant to whether accumulation of K27me3 in the cKO impacts DEG and therefore may play a compensatory role. However, additional analysis could more decisively test the hypothesis. For example, do non-dysregulated genes containing DMRs show less K27me3 accumulation than down-regulated genes, but more K27me3 accumulation than upregulated genes? The authors could include a boxplot of a well-matched set of non-DE genes with overlapping DMRs in panel 6E to test this. A more powerful analysis, however, might be to examine a large set of genes from the genome that are selected to contain similar numbers of DMRs (in order to control for relative potential effects of demethylation) and ask if changes in K27me3 correspond with up-regulation (low K27me3 accumulation), no dysregulation (intermediate accumulation) or downregulation (high K27me3 accumulation).

We apologize for the confusion. We have added a panel using the same method used in Figure 5B (see “Enrichment test of DMRs and other genomic regions” in the Materials and methods section) to show the odds ratio of DMRs overlapping the H3K27me3 DM regions (now in Figure 4 — figure supplement 3D, right panel).

In addition, in the previous version of our manuscript, the enrichment of DMRs in DM regions was also assessed in the Venn diagram in Figure 4 — figure supplement 3D, and we have corrected the figure reference in the main text and the figure legend for Figure 4 — figure supplement 3. We tested this using the `fisher` sub-commands in the BEDTools Suite, where we checked if the amount of overlap between the 2 sets of regions is more than expected given their coverage and the size of the genome. This is in most cases analytically the same as shuffling the genome and checking the simulated (shuffled) versus the observed.

As the reviewer suggested, we have selected a more stringent set of regions as control. For each DM H3K27me3 peak, one non-DM H3K27me3 peak with a similar peak size and similar H3K27me3 signal in the control (wild-type) mice was selected as the matched control region using the greedy nearest neighbor matching in the R package `MatchIt`. Figure 4 — figure supplement 3C shows the distribution of H3K27me3 signal in the DM and non-DM peak sets:

We then counted the number of overlaps with DMRs in these matched non-DM peaks and compared those with the observed DM peaks (Figure 4 — figure supplement 3D, left panel). We found that the DM-peaks have significantly more overlaps with DMRs compared to the non-DM peaks (Fisher exact test p < 2.2e-16, odds ratio OR=2.70):

We have added these additional figures in Figure 4 — figure supplement 3 and described these results as follows in the main text:

“Likewise, P39 cKO DMRs were significantly enriched in peaks (22.4% overlapped H3K27me3 peaks in cKO, p<1e-100, Figure 5B) and DM regions of H3K27me3 (Figure 4 — figure supplement 3D). The DM regions of H3K27me3 had significantly more overlaps with DMRs compared to the non-DM control regions (Fisher exact test p < 2.2e-16, OR: 2.70, Figure 4 — figure supplement 3C-D).”

The methods used here are clarified in the Material and Methods section as follows:

“To select a set of non-DM control regions to match the base levels of H3K27me3 in DM regions, we started with the union peaks of the control and cKO samples and removed any peaks that overlap the DM regions. From these non-DM peaks, we used the R package MatchIt (Ho et al., 2011) (v3.0.2) to select regions with matched peak lengths and H3K27me3 levels (in the unit of RPKM) as those in the DM regions, with the greedy nearest neighbor matching using logistic link propensity score as a distance measure (requiring the standard deviation of the distance to be less than 0.01).”

b. On p19 the authors state that "P39 cKO DMRs were significantly enriched in peaks and DM regions of H3K27me3 (28.1% overlapped H3K27me3 peaks in cKO, p<1e-100) (Figure 6B)." However, 6B only presents the overlap with peaks in each genotype as an odds ratio and doesn't make it clear how the enrichment of DMRs in DM regions was assessed. This appears to be assessed to some degree in S8F (note the S8 figure legend appears to be wrong for this panel), but no analysis is presented to control for the general tendency of DMRs and H3K27me3 peaks to overlap. It seems imperative, given that DMRs already overlap with H3K27me3 peaks, that a resampling approach is used comparing DM peaks to non-DM K27me3 peaks that are matched to have similar wild-type H3K27me3 signal and size as DM peaks. Likewise, what would the changes in H3K27me3 signal presented in 6C look like at non-DMR regions that contain similar baseline wild-type H3K27me3 signal? Without these analyses it is not clear that DMRs are particularly unique in showing increased K27me3 signal.

As for the reviewer’s suggestion of looking at non-DMR regions that contain similar baseline wild-type H3K27me3, it will be computationally intense to search for such matching regions as non-DMR regions are essentially the complement of DMR regions from the whole genome. And it will take time and luck to shuffle the DMR regions, compute the baseline H3K27me3 signal and get a match to the observed DMR regions. Instead, we addressed this comment with the following two approaches.

First, in Figure 5 — figure supplement 1A we analyze the ChIP-seq signal enrichment for two types of shuffle control regions. The first group of shuffle regions (dotted lines) is generated by shuffling the DMRs randomly across the same chromosome (excluding blacklist regions). These regions represent random background noise. Indeed, we find that the ChIP-seq signal is flat for these control regions in every sample and every histone mark. The second group of shuffle regions (dashed lines) is selected to meet the same criteria as the first group with an additional restriction that the shuffles must reside within the P39 H3K27me3 peak regions (the union of P39 Control and P39 cKO H3K27me3 peaks). These regions should at least have some P39 H3K27me3 signal, and as seen in the figure when averaged they formed a line that peaks at the center in the P39 H3K27me3 samples. Even though on average they have a higher enrichment of H3K27me3 compared to the observed DMRs, the differences between the cKO and Control are much smaller than those in the observed DMRs.

Secondly, we tiled the genome with 1kb bins and grouped them into bins with overlapping DMRs and bins without overlapping DMRs. For these two groups of bins, we plotted the mean changes of H3K27me3 in cKO vs. Control as a function of the baseline H3K27me3 signal in the control animals, stratified by the deciles of the baseline H3K27me3 signal (Figure 5 — figure supplement 1B). To avoid double-dipping (using the control signal in both the x-axis and y-axis), we used the baseline signal from the Control replicate 1 on the x-axis and the baseline signal from the Control replicate 2 on the y axis, and vice versa. In both cases, we observed bigger increases in H3K27me3 in bins with overlapping DMRs (Wilcoxon rank-sum test p < 0.001 in each bin), even after controlling for the baseline H3K27me3 level:

These new analyses indicate that DMRs are especially enriched in regions of increased H3K27me3. We have added these figures in Figure 5 — figure supplement 1A-B and described these results in the main text as follows:

“Moreover, H3K27me3 was more abundant at the center of DMRs in cKO compared to control neurons at P39 (∆ = 0.15 in the unit of fold enrichment vs. IgG, 5.04% increases, Figure 5C). There were much smaller changes of H3K27me3 at these DMRs in newborn (P0, ∆ = 0.0075, 0.27% changes) or fetal (E14, ∆ = -0.028, -1.02% changes) neurons (Figure 5C), and the increases were not seen in randomly shuffled regions (Figure 5 — figure supplement 1A). The signal of H3K4me3 and H3K27ac at the DMRs was also elevated in the cKO, to a lesser extent (H3K4me3: ∆ = 0.068, 2.92% changes; H3K27ac: ∆ = 0.13, 4.64% changes). … When accessing the H3K27me3 changes as a function of baseline H3K27me3 levels in the control across the genome tiled in 1kb bins, we observed bigger H3K27me3 increases in bins with overlapping DMRs compared to bins without overlapping DMRs (Wilcoxon rank-sum test p < 0.001 in each bin, Figure 5 — figure supplement 1B). These results indicate that DMRs are particularly unique in showing increased H3K37me3 signal.”

c. The increase in H3K27me3 in the cKO at the mab21l2 locus at P0 (figure 5B), before methylation has or has not been established, suggests that H3K27me3 may change before methylation differences, rather than in reaction to changes in methylation. Is this a common effect seen across other genes? How does this impact the idea that compensation occurs due to lack of methylation? Related to this issue, on p.19 no specific plot is referenced to support the statement that "There was no corresponding increase of H3K27me3 at these DMRs in newborn (P0) or fetal (E14) neurons." A quantitative analysis as shown in 6C but for the E14 and P0 time-points could assess this directly.

We apologize for the confusion. At the *Mab21l2* locus (now in Figure 4B) there is no significant increase of H3K27me3 at P0 or E14 (i.e. there is no horizontal red bar under the P0 and E14 tracks; and our DiffBind analysis used a threshold of FDR<0.05). We have added a shaded box to highlight the region with significant increases of H3K27me3 at P39 to better illustrate our finding here (Also see our replies to reviewer #1, comment 3 for the quantification of the H3K27me3 levels at this locus).

We agree with the reviewer that quantitative analysis is needed to support our statement regarding the lack of increased H3K27me3 at DMRs at P0 and E14. Indeed, this analysis was included (now in Figure 5C) and we have updated the text to directly reference this figure panel as follows:

“Moreover, H3K27me3 was more abundant at the center of DMRs in cKO compared to control neurons at P39 (∆ = 0.15 in the unit of fold enrichment vs. IgG, 5.04% increases, Figure 5C). There were much smaller changes of H3K27me3 at these DMRs in newborn (P0, ∆ = 0.0075, 0.27% changes) or fetal (E14, ∆ = -0.028, -1.02% changes) neurons (Figure 5C), and the increases were not seen in randomly shuffled regions (Figure 5 — figure supplement 1A).”

Finally, we have also shown the H3K27me3 changes did not happen in earlier timepoint as we found almost no significant differentially modified H3K27me3 regions in E14 or P0 (Figure 4 — figure supplement 1D). Also, if we focus on the 4,040 significant up-regulated H3K27me3 regions found in P39, the increased signal changes in cKO vs. Control in P39 are not correlated with those in E14 or P0 (Figure 4 — figure supplement 2B and D).

d. Why were developmental DMs called between E14 and P39 while DMRs were called between P0 and P39? Developmental comparisons would be more justified if both used P0 and P39.

Our choice to analyze the DMs between E14 and P39 was motivated by several characteristics of our dataset. Unfortunately, we do not have methylation data for E14 excitatory neurons, therefore developmental DMRs could only be called between P0 and P39. Moreover, we found much more significant developmental DM H3K27me3 regions in P39 vs. E14 compared to P39 vs. P0 (Figure 6 — figure supplement 1A). We believe that one of the reasons for this is that the E14 ChIPseq data are of better quality than the perinatal data from P0 (Figure 6 — figure supplement 2). For this reason, in our original submission, we chose to use E14 and P39 to call developmental H3K27me DMs. We now provide a new figure for the reviewers (Figure R2) to show corresponding results for Figure 6B-E and Figure 6 — figure supplement 1C-G using the set of developmental H3K27me3 DMs between P39 and P0. In these figures, we demonstrate that the conclusions drawn in the previous version are verified in the P39 vs. P0 comparison.

e. In the absence of additional data supporting H3K27me3 accumulation as a functional compensation mechanism for loss of DNA methylation, the title of Figure 6 should not so strongly claim that H3K27me3 changes compensate for mC loss.

We have revised the title of this figure (now Figure 5): Increased H3K27me3 correlates with the loss of postnatal DNA methylation.

3. While initially assessed by the authors, mCH changes are not integrated into the analysis of H3K27me3 changes. Previous studies by this group and others have suggested the importance of mCH in gene regulation within specific neuron populations and, given the interest to the field, more thorough analysis is warranted regarding this neuronal methylation and H3K27me3.a. Given the association, even if limited, between differential mCH loss and gene dysregulation in the cKO, it is relevant to examine how changes in K27me3 relate to, or compensate for, loss of mCH. For example, do genes that are not dysregulated in the cKO, and normally have high mCH (which is lost in the cKO) show higher increases in K27me3 compared to non-dysregulated genes that have low mCH (and therefore lose less mCH)? Reciprocally, one could ask if a population of genes selected to have similar, high mCH loss in the cKO, show differential gene dysregulation as a function of differential K27me3 accumulation.

We appreciate the reviewer’s insightful suggestions for analysis of the relationship between the loss of mCH and the changes in the H3K27me3 signal. We agree that since the loss of mCH and the loss of mCG are highly correlated (Figure 3 — figure supplement 3D), we would expect to see to a certain extent a similar compensation of H3K27me3 in genes/regions with loss of mCH. We have added a figure to Figure 5 — figure supplement 2B to show the changes in the H3K27me3 signal as a function of the loss of gene body mCH in non-DE genes (stratified by the deciles of δ mCH):

We indeed observed a larger increase of H3K27me3 signal in the gene body of genes that lost most mCH in the cKO (genes that have high mCH in the control; the leftmost decile in the plot), compared to genes that lost less mCH (genes that normally have low mCH).

In addition, we grouped the genes by the quantiles of loss of mCH in cKO and plotted the gene expression fold-changes as a function of the changes of H3K27me3 in gene body for each of these gene sets (Figure 5 — figure supplement 2C):

This shows that genes that lost the most mCH in the cKO (genes that have high mCH in the control; red line) tend to have increased expression in the cKO, compared to genes that lost the least mCH (genes that have low mCH to begin with; blue line), which is consistent with our results in Figure 3E. Moreover, we can see a negative trend between the changes of gene body H3K27me3 signal and gene expression fold-change in the gene set with the highest baseline mCH, in which genes that showed the biggest increases of H3K27me3 were inclined to decrease in expression whereas genes that had small increases or decreases of H3K27me3 signal were generally up-regulated in gene expression. Such correlation was not observed in genes with the lowest baseline mCH.

We have added these plots in Figure 5 — figure supplement 2B-C and described the results in the main text as follows:

“We next analyzed how changes in H3K27me3 related to the loss of mCH. In all non-DE genes (FDR ≥ 0.05), we observed a larger increase of H3K27me3 signal in the gene body of genes that lost most mCH in the cKO (Figure 5 — figure supplement 2B). Indeed, when we grouped all genes by the extent of the loss of mCH, we found that genes that lost the most mCH had a negative correlation between the changes in gene body H3K27me3 signal and gene expression fold-change. Such correlation was not observed in genes that did not lose mCH (Figure 5 — figure supplement 2C). These results suggest that H3K27me3 may have a role in repressing expression specifically in regions that lose repressive non-CG DNA methylation.”

b. What are the changes in mCH at DM H3K27me3 sites? Since mCG and mCH loss are correlated, one might expect that these are sites that normally also contain high mCH. It might not be feasible to call mCH DMRs given the technical limitations, but quantification of this signal can be done at DMs and DMRs, and the potential for it to contribute to these effects could be discussed.

As suggested by the reviewer, we have estimated the changes of mCG and mCH in the DM H3K27me3 regions and also compared them with the changes in matched non-DM regions (see our response to comment 2b for the methods we used to select these matched non-DM regions). We found that the DM regions showed significantly larger decreases in both mCG and mCH when compared to non-DM regions (Wilcoxon rank-sum test p < 0.0001).

We have added these plots in Figure 4 — figure supplement 3E and described the results in the main text as follows:

The DM regions also have larger decreases in both mCG and mCH when compared to non-DM regions (Wilcoxon rank-sum test p < 0.0001, Figure 4 — figure supplement 3E).4. The authors present evidence of PRC2 binding at DEGs (Figure 5A) as a motivation to assess H3K27me3 in their mice. However, they then observe changes in H3K27me3 that they contend do not occur under normal conditions and propose that these changes are an underlying mechanism that blocks gene dysregulation in these mice. How are we to reconcile this original rationale of a normal function for PRC2 at dysregulated genes with the model of de novo compensatory H3K27me3 accumulation? Some explanation by the authors of how these separate notions may relate to one another would be helpful.

The BART results for up-regulated and down-regulated DE genes (now in Figure 4A) were bioinformatics predictions that drove us to look into the Polycomb repression via H3K27me3 profiling. BART predictions are based on ChIP-seq data in various cell lines and tissues from the ENCODE project. This means that the potential PRC2 binding at our DEGs may normally happen in systems other than the brain or pyramidal neurons, or at other time points during development. Our results showed that these DEGs were normally regulated by DNA methylation in the pyramidal neurons but they can still be regulated by PRC2 in the current system if the DNA methylation was abolished.

We have added these explanations in the Discussion section of the manuscript as follows:

“Overall, our results suggest that when DNA methylation is disrupted, H3K27me3 might partially compensate for the loss of mCG and/or mCH and act as an alternative mode of epigenetic repression. Nevertheless, we did not find differential expression in any of the four core components of PRC2 (Ezh2, Suz12, Eed and Rbbp4) in adult Dnmt3a cKO animals. It is possible that the increased H3K27me3 was mediated by transient expression of PRC2 components during development in the cKO. Furthermore, the predictions from BART (Figure 4A) were derived from various cell lines and tissues from the ENCODE project (Davis et al., 2018; ENCODE Project Consortium, 2012), suggesting that the potential PRC2 binding at our DEGs may normally happen in systems other than the brain or pyramidal neurons, or at other time points during development. Additional experiments which directly manipulate components of the PRC2 system are required to further test the potential compensation mechanism.”

5. Some effects on H3K27ac are observed at DMRs, but the authors ignore the prospect that these effects could drive changes in gene expression. Previous groups have shown significant changes in H3K27ac in DNMT3A models, and some of the data the authors present suggest that this could also explain some of the observations in their model.a. While DM K27ac peaks are not identified in the cKO, in figure 6B there is a similar enrichment of DMRs to associate with K27ac peaks in the cKO vs control as there is for K27me3. In addition, there is a similar increase in odds ratio of enrichment of K27ac peaks to the of K27me3 at DMRs for the cKO in figure 6B. The analysis in S8F-G that is referenced when referring to K27ac is either missing (S8G) or is not showing data for K27ac (S8F). The authors should at least comment in the Results section that these signals exist and may be significant for gene regulation.

In the new set of DMRs from the new batch of DNA methylation data (see responses to the editor on why we introduced a new batch of sequencing data), we didn’t see the enrichment of overlapping DMRs in H3K27ac (now in Figure 5B). But indeed, we still see the enrichment of the P39 H3K27ac signal in the center of DMRs (Figure 5C). We decided to focus on the changes of H3K27me3 since it’s more evident, but we agree with the reviewer that H3K27ac may also have an impact on regulating the expression of the DEGs. We apologize for the confusion in referring to Figure S8F-G, and we have corrected the text and added some comments regarding H3K27ac as follows:

“The signal of H3K4me3 and H3K27ac at the DMRs was also elevated in the cKO, but to a lesser extent (H3K4me3: ∆ = 0.068, 2.92% changes; H3K27ac: ∆ = 0.13, 4.64% changes).”b. Are two biological replicates sensitive enough to call differences in H3K27ac? In the context of a small number of replicates and the subtle magnitude of changes in gene expression that are observed, negative results for differential peaks should be interpreted with caution. In contrast, the larger peaks observed in K27me3 may have led to increased ability to call differential peaks.

We agree with the reviewer that our power for discovering changes in H3K27ac is somewhat limited by the sample size of two replicates per condition. We have added some comments to remind the readers of this potential issue in the Discussion section as follows:

“Indeed, we found that the Dnmt3a cKO DMRs overlapped with regions that gain methylation during normal postnatal development (Figure 3G-H). These DMRs had increased H3K27ac in the Dnmt3a cKO (Figure 5C), consistent with observations in adult animals lacking Dnmt3a specifically in GABAergic Sst- or Vip-expressing interneurons (Stroud et al., 2020). In those experiments, embryonic gene-regulatory elements had lower cytosine methylation and increased H3K27ac and H3K4me1 (Stroud et al., 2020). This suggests an essential role for Dnmt3a and DNA methylation in shaping the transcriptome during development in part via the inactivation of embryonic enhancers, with potentially long-lasting effects on the gene expression pattern of mature neurons. Although we did not detect individual peaks with a statistically significant difference in H3K27ac between cKO and control (Figure 4 — figure supplement 1C), this could be due to a small number of replicates and the subtle magnitude of changes in gene expression relative to the large number of tested peaks. Future experiments using more replicates or orthogonal techniques like CUT&RUN and CUT&Tag could help to further investigate the relationship between the Dnmt3a-dependent DNA methylation and the activation of enhancers.”

6. Though the characterization of DNMT3A cKO in excitatory neurons is novel, if the authors are claiming that these neurons specifically are driving behavioral phenotypes then further contextualization and characterization of the extent of conditional deletion is needed.a. It would be useful to justify the rationale for investigating excitatory neurons, and how using this class of neurons can uncover novel findings. Existing studies of DNMT3A cKO have shown behavioral and epigenomic effects in pan-neuronal and inhibitory neuron conditional knockouts. Comparisons between these models and discussion of differences could offer strong insights into what unique circuits may be most disrupted by DNMT3A cKO in excitatory neurons.

Our previous studies (e.g. Lister et al., 2013; Mo et al. 2015; Luo et al. 2017) have shown that CG and non-CG DNA methylation across all neuron types has a highly cell type specific distribution that is globally remapped during postnatal brain development. To understand the functional significance of this epigenetic regulation, we chose to focus on the major neuronal populations with established roles in regulating cognitive behavior. Our previous study (Lavery et al., 2020) analyzed the loss of Dnmt3a in GABAergic neurons. Here, we chose to focus on the other major neuronal population, the excitatory (glutamatergic) neurons in the frontal cortex.

Note that both pan-neuronal and pan-GABAergic cKOs (e.g. (Feng et al., 2010; Morris et al., 2014; Nguyen et al., 2007)) affect many more regions of the brain than what is affected in our study, thus leading to confounding compensatory/deleterious phenotypes. Previous studies using those cKOs had profound phenotypes and died before reaching early adulthood. Because we wanted to analyze the effect of blunting the increase in *Dnmt3a* during the perinatal period in a cell-specific manner, without killing the mice, we believe our approach has allowed for a much more detailed study of the regulation of neuronal DNA methylation. Although the phenotypes observed in the present cKO are mild, both at the functional and transcriptional level, we still found significant changes that are more reminiscent of those found in neurodevelopmental disorders. *Dnmt3a* de novo mutations have been linked to autism spectrum and other developmental disorders. Our present results may point to the period of gain of methylation in postmitotic excitatory neurons as a critical period in the development of these disorders.

b. If the authors are claiming that behaviors and phenotypes are the result of specifically knocking out DNMT3A in all excitatory neurons, further evidence should be provided that the Cre-line is being expressed sufficiently and specifically in excitatory neurons, and that their genomic data is faithfully capturing the majority of excitatory cells while excluding non-excitatory cells.

We thank the reviewer for raising this question. The Neurod6-Cre mouse line (also referred to as NEX-Cre) used in our study is a knock-in mouse line that expresses Cre recombinase under the control of Neurod6 regulatory sequences (Goebbels et al., 2006). Neurod6 is a transcription factor exclusively expressed in the central nervous system – most prominently in excitatory neurons of the neocortex and hippocampus (e.g. see Yao et al., *Cell* 2021), and its expression parallels excitatory neuronal differentiation and synaptogenesis during brain development (see Figure 1 — figure supplement 1B).

Consistent with the expression of NeuroD6, Goebbels et al. (2006) observed the most prominent Cre activity in the neocortex and hippocampus, starting from around embryonic day 11.5. Moreover, Cre-mediated recombination marked pyramidal neurons and dentate gyrus mossy and granule cells and was absent from proliferating neural precursors of the ventricular zone, interneurons, oligodendrocytes, and astrocytes (Goebbels et al., 2006). We confirmed that Neurod6-dependent Cre recombination occurred only in excitatory neurons using the INTACT mouse (Mo et al., 2005), in which Cre-mediated excision of the transcription stop signals activates the expression of the nuclear membrane tag (Sun1 -sfGFP-myc) in the cell type of interest. Since recombination only occurs in excitatory neurons, we expect excision of exon 19 from *Dnmt3a* will only occur in those neurons expressing Cre recombinase. Figure 1 — figure supplement 2A-C depicts the staining of Sun1-GFP across the brain in non-inhibitory Neurod6+ neurons, confirming previous results (Goebbles et al., 2006). Note that the sfGFP-tag decorates the nucleus of cells where recombination occurs.

We also added the following Figure 1 — figure supplement 2D to show the expression of pan-neuronal, excitatory neuron and inhibitory neuron marker genes in our RNA-seq data, in which we observed high expression of excitatory neuron markers and very low expression of inhibitory neuron markers, further demonstrating that the neurons carrying the GFP label belong to the excitatory neuron population.

7. The manuscript would be further improved by clarification of some methods and statistics, and modifications in data presentation:a. The authors should provide information on the statistics performed on the behavior data. Are there significant differences between sexes, or significant sex by genotype interactions for these behavioral tasks? A two-way ANOVA using genotype and sex would be useful in understanding the impact of these variables. There appears to be many effects that are sex-specific, yet this appears to be overlooked in the text. Furthermore, if there are robust sex-specific effects then the subsequent molecular and genomic analyses should be done using both sexes, and comparisons could be done to explain the molecular underpinnings of these differences.

We thank the reviewer for raising this point, for which we would like to provide clarification. The experiments in this section (Figures 1, Figure 1 — figure supplement 4 and 5) were first and foremost designed to investigate whether the loss of *Dnmt3a-*dependent DNA methylation during the maturation of pyramidal cells might lead to endophenotypic markers of neurodevelopmental illnesses. Male mice were used in generating all genomic datasets. We performed behavioral experiments in cohorts of male mice, which led to the discovery of impairments in social behavior, PPI, and working memory. We subsequently investigated if those same behaviors were affected in a cohort of female mice, finding that the loss of *Dnmt3a* caused similar impairments in working memory, but not in PPI or social behavior. Because the male and female cohorts were tested in completely independent experiments, it would not be appropriate to perform a two-way ANOVA using sex as a variable, as this could result in an inaccurate depiction of results. Yet, the wider range of behavioral impairments identified in the male cohort is noteworthy, as it correlates well with human neurodevelopmental disorders, such as schizophrenia and autism, in which males are disproportionately more affected than females. We have edited the Discussion to clarify that even though it was outside the scope of the present work, our data provide support for designing a comprehensive follow up study aiming to dissect sex-specific effects of *Dnmt3a* and their underlying causes.

b. The authors might also consider showing preference index when analyzing social approach data when comparing genotypes.

We thank the reviewer for this suggestion, which we have carefully considered. We decided to keep our results as raw "time spent on side" to take into account the differing time spent in the empty space, which showed high significance and would be lost if the results were reported as preference index. We edited the results text in order to mention this finding clearly:

“Moreover, when tested in a three-chamber box in which one of the sides contained a novel mouse and the opposite a novel object, male Dnmt3a cKO animals spent less time in the former, opting, instead, to remain significantly longer in the center (empty compartment), which is suggestive of reduced exploration and social interest (Figure 1D, left panel; p = 0.01048).”

c. What is the background set of genes for 2B-C? Synaptic genes are enriched in both up- and down-regulated gene lists. How is gene expression normalized? Is there a discovery bias for more highly expressed genes?

In the previous version of our manuscript, we used expressed genes in P39 as the background set in the GO analysis for the DE genes. Expressed genes were defined as genes with CPM (counts per million) greater than 2 in at least two samples. In the new batch of RNA-seq data (see responses to the editor on why we introduced a new batch of sequencing data), we identified fewer significant DE genes and hence found no significant enrichments in any GO terms. Therefore, we have removed all the figures regarding the GO analysis for the DE genes.

d. What value was used to identify equally expressed unchanged genes in figure 4B, TPM? Identifying equally expressed genes can be complicated for nuclear RNA where unprocessed RNAs contribute differentially to total counts based on the length of the gene. In addition, previous analysis of mCH enrichment at dysregulated genes in inhibitory neuron-specific KO of DNMT3A also found DEGs to be differentially methylated based on direction of change but showed that all other genes were more lowly methylated than these two groups (Lavery et al. 2020). Here the authors see the same relationship between DEG groups but find other genes to be higher in mCH rather than lower. Do the authors have any perspective on why these differences are observed?

We used the R package `MatchIt` to select non-DE genes with matched expression levels in the unit of TPM. We have added Figure 3 — figure supplement 4B to show these non-DE genes have similar expression levels as the DE genes in the P39 Control samples, and their expressions were not changed in the P39 cKO samples.

With the new batch of RNA-seq and MethylC-seq data with bona fide biological replicates (see responses to the editor on why we introduced a new batch of sequencing data), we updated the mCH profile in the DEGs and corresponding non-DE control genes (now in Figure 3D):

We now see that the up-regulated genes have higher mCH than the downregulated genes, and non-DE genes have lower mCH than the up-regulated genes but also indistinguishable levels from the down-regulated genes. This in general agrees with the findings in Lavery et al. We presumed the discrepancy between the findings in Lavery et al. and our previous version of the manuscript might be due to the differences in the power of detecting DE genes, the choice of thresholding significant DEGs, the method of selecting non-DE control genes, and probably some unknown distinctions of biological mechanism in regulating gene expression after *Dnmt3a* cKO in inhibitory neurons vs. in excitatory neurons.

e. On p22 the authors state "We found that developmental peaks (DevLoss and DevGain) were relatively unaffected by the cKO(H3K27me3 = 0.11, -0.20 respectively, in units of log10(CPM+1)), whereas Group cKO had ten-fold larger mean effect (H3K27me3 = 1.12 respectively)." Given the y scales in figures 7E and S10H it is unclear how they derived a ∆ value of 1.12 on a log 10 scale for Group cKO (e.g. in S10H the median values look much less 1 unit different on a log10 scale). Please clarify the actual magnitude of effect sizes.

We are sorry for the confusion. In the main figure (now in Figure 6C-E) we normalized the H3K27me3 signal to the means of the two P39 Control samples and plotted the fold changes. In the said supplemental figures (now in Figure 6 — figure supplement 1E-G) we plotted the H3K27me3 signal in units of log10(CPM+1) for each sample. In the previous version of our manuscript, the δ values were achieved by running a paired t-test on the raw CPM scale. To avoid confusion, we have rerun the paired t-test on the log10(CPM+1) scale and updated the δ values in Figure 6 — figure supplement 1E-G. We have also edited the relevant statements in the main text as follows:

“To further stratify the joint distribution of developmental and Dnmt3a cKOdependent changes in both H3K27me3 and DNA methylation, we assigned peaks to three groups (Figure 6C-E and Figure 6 — figure supplement 1EG). “Group DevLoss” and “Group DevGain” peaks lose or gain H3K27me3 during development, respectively (Figure 6C-D and Figure 6 — figure supplement 1E-F). “Group cKO” peaks have higher H3K27me3 in the Dnmt3a cKO compared to control at P39 (Figure 6E and Figure 6 — figure supplement 1G). We found that developmental peaks (DevLoss and DevGain) were relatively unaffected by the cKO (H3K27me3 = 0.02, -0.03 respectively, in units of log10(CPM+1)), whereas Group cKO had 4.5-fold larger mean effect (H3K27me3 = 0.09 respectively). Group cKO peaks also experienced greater loss of mCG (mCG = -23.5%) than Group DevLoss (13.0%) or DevGain (18.6%) (Figure 6C-E and Figure 6 — figure supplement 1E-G, middle and right panels). These results suggest that regions prone to alteration of H3K27me3 by Dnmt3a cKO are distinct from the regions affected by developmentally dynamic H3K27me3.”

f. The authors describe the association between gene body mCH and gene expression changes as only accounting for 1% of the variance. It would be useful to have a point of reference to understand if this is a low value or a high value in comparison to other potential predictors of gene expression changes. For example, how much does the proposed overcompensation by H3K27me3 accumulation contribute to the genome-wide variance in gene expression? In addition, is it the case that variance in the measured fold-changes will be mitigated by addition of more replicates to more accurately estimate true fold-changes? If this is the case, it should be noted.

Thank you for the valuable suggestion to provide a point of reference for the magnitude of the correlation between mCH and gene expression. The variance explained (R-squared) between the gene body mCH and RNA log fold-change was 0.46%. To give a quantitative context for this, we estimated the total explainable variance as the R-squared between RNA log fold-change from independent biological replicates. That is, we selected one of the cKO and one control sample to calculate the first replicate logFC value; we then used the second cKO and second control sample to get a second, independent replicate logFC estimate. We found that the correlation between these was 1.30%. This shows that, although the mCH R-squared is small, it is nevertheless a significant fraction of the total explainable variance given the noise in our RNA-seq data. As the reviewer suggests, it is likely that adding more data (more biological replicates) would reduce the noise in both the RNA logFC and the mCH measurements, which might increase the explained variance. We have revised the Results as follows:

“The difference in gene body methylation (cKO – Control) was negatively correlated with gene expression changes, consistent with repressive regulation (Gabel et al., 2015; Lavery et al., 2020) (Figure 3E). This correlation accounted for 0.46% of the variance of differential gene expression, whereas the total explainable variance (R2 between biological replicates) was 1.30% (Figure 3 — figure supplement 4D). The strength of the association between mCH and mRNA changes may be limited by the use of only two biological replicates in our dataset.”

Furthermore, we used a linear model to test how much the δ H3K27me3 signal may contribute to the variance of RNA logFC between cKO and control in 14,755 expressed genes. We found that the R-squared value is 0.0599%, which is smaller than the R-squared for mCH (0.456%, see Figure 3 — figure supplement 4D).

g. Using different color schemes would be helpful in 8A and 8B to avoid confusion.

We have modified the said figure (now in Figure 7A-B) as suggested to avoid confusion.

h. How was clustering analysis done in 8D? What do the values for K27me3 Diff. Mod and DEG enrichment indicate (i.e. what does the scale from 0 to 0.02 represent)?

The clustering of DMVs (now in Figure 7C) was originally performed using k-means. In the resubmission, we have updated this analysis to make the results more evident, and as a result we modified the clustering procedure. The new procedure sorts DMVs according to the following criteria: (1) whether they overlap an H3K27me3 DM region; whether they overlap an up- or down-regulated DE gene (with absolute log2(Fold-change)>0.2); mean mCG>0.3 in P39 Control, P39 cKO; P0 Control, or P0 cKO; and finally by the average level of H3K27me3, H3K3me3 and H3K27ac.

The “Fraction DM H3K27me3” shows what fraction of the length of the DMV overlaps a DM H3K27me3 region (called using diffbind).

We now show directly the mean mRNA logFC for all genes contained within each DMV.

We have added the description of this analysis in the method section as follows:

“For the clustering visualization in Figure 7C, we sorted DMVs according to the following criteria: (1) whether they overlap an H3K27me3 DM region; whether they overlap an up- or down-regulated DE gene (with absolute log2(Fold-change)>0.2); mean mCG>0.3 in P39 Control, P39 cKO; P0 Control, or P0 cKO; and finally, by the average level of H3K27me3, H3K3me3 and H3K27ac. The “Fraction DM H3K27me3” shows what fraction of the length of the DMV overlaps a DM H3K27me3 region (called using DiffBind). And we also plotted the mean mRNA logFC for all genes contained within each DMV.”

Reviewer #3 (Recommendations for the authors):– The authors delete exon 19 of Dnmt3a. I think they should provide a more thorough description of what this deletion is expected to cause: loss of function due to premature stop codon, loss of enzymatic activity, etc. Moreover, it would be desirable to show by immunostaining whether DNMT3A protein is specifically lost in excitatory neurons.

We thank the reviewer for raising this question and allowing us to clarify an important point. Deletion of exon 19 in the mouse *Dnmt3a* gene is expected to produce a deletion of 50 amino acids (aa 623-673) in the methyltransferase domain (encompassing the S-adenosyl-L-methionine binding site), thus leading to disruption of the catalytic activity of the enzyme (Lyko 2018). In addition, this approach resulted in a marked downregulation of *Dnmt3a* protein levels, as quantified by western blot experiments (Figure 1 — figure supplement 3B). Such a downregulation in *Dnmt3a* expression was matched by an almost complete loss of mCH and loss of mCG remethylation in excitatory neurons confirming the lack of activity of Dnmt3a in excitatory neurons from frontal cortex (Figure 3C). Please also see our response to Reviewer #2 comment 6b where we performed triple labeling of brain slices, showing that the Cre-dependent recombination, and thus the deletion of *Dnmt3a*, occurs in excitatory, but not inhibitory, neurons.

– Regarding the DMRs identified in DNMT3A KO neurons, the authors should use the generated profiles for H3K27ac in P0 and P39 WT neurons to determine which DMRs overlap enhancers that are active in any of these two neuronal stages. This is more relevant than overlaps with enhancers that could be active in any given cell type. Most importantly, the authors should evaluate the impact the loss of DNA methylation can have on enhancer activity by measuring H3K27ac levels and eRNAs in DNMT3A KO neurons. Measuring eRNA levels can be particularly relevant, as recent evidences suggest that is a better marker of enhancer activity than H3K27ac.

We agree that it is important to analyze the impact of the loss of DNA methylation on enhancer activity. For P39 samples, we analyzed the overlap of H3K27ac peaks in pyramidal neurons with DMRs (Figure 5A-B and Figure 5 — figure supplement 1A). Moreover, in Figure 5C we plotted the mean level of H3K27ac in the region around DMRs. These analyses show that enhancer activity is indeed increased in regions that lose DNA methylation in the cKO. Please also see our responses to Reviewer #1 comment 3 and Reviewer #2 comment 5 regarding the analyses on H3K27ac.

We would like to clarify that we did not perform H3K27ac ChIP-seq for P0, so we were unable to specifically examine the association of enhancer activity at the perinatal timepoint.

We agree that it would be interesting to further examine the enhancers using eRNAs, however, we found that the depth of coverage of our RNA-seq data was not sufficient to address this.

– The authors should aim at performing experiments that address the proposed compensatory role for PRC2 and H3K27me3. I understand that generating a KO for PRC2 could be rather complicated, but perhaps the authors could consider using new drugs that act as very specific inhibitors against the enzymatic activity of PRC2.

We appreciate the reviewer’s suggestion, and we agree that it will need further experiments to directly test the hypothesis of the compensatory role for PRC2 and H3K27me3. But such experiments are out of the scope of this manuscript. We have modified the text to weaken our claims regarding the compensation effect of H3K27me3.